# NODE2KET: EFFICIENT HIGH-DIMENSIONAL NETWORK EMBEDDING IN QUANTUM HILBERT SPACE

**Hao Xiong, Yehui Tang, Yunlin He, Wei Tan, Junchi Yan**[*]
Department of Computer Science and Engineering, Shanghai Jiao Tong University
{taxuexh,yehuitang,anonimit_y,wilson.t,yanjunchi}@sjtu.edu.cn
https://github.com/ShawXh/node2ket

## ABSTRACT

Network embedding (NE) is a prominent technique for network analysis where the nodes are represented as vectorized embeddings in a continuous space. Existing works tend to resort to the low-dimensional embedding space for efficiency and less risk of over-fitting. In this paper, we explore a new NE paradigm whose embedding dimension goes exponentially high w.r.t. the number of parameters, yet being very efficient and effective. Specifically, the node embeddings are represented as product states that lie in a super high-dimensional (e.g. $2^{32}$-dim) quantum Hilbert space, with a carefully designed optimization approach to guarantee the robustness to work in different scenarios. In the experiments, we show diverse virtues of our methods, including but not limited to: the overwhelming performance on downstream tasks against conventional low-dimensional NE baselines with the similar amount of computing resources, the super high efficiency for a fixed low embedding dimension (e.g. 512) with less than 1/200 memory usage, the robustness when equipped with different objectives and sampling strategies as a fundamental tool for future NE research. As a relatively unexplored topic in literature, the high-dimensional NE paradigm is demonstrated effective both experimentally and theoretically.

*Hilbert space is a big place!*

– Carlton M. Caves, cited in *Quantum Computation and Quantum Information*,
10th Anniversary Edition. Massachusetts Institute of Technology, 2010.

## 1 INTRODUCTION

Network embedding (NE), has been extensively studied for decades. By representing nodes in networks as semantically informative embeddings that are mathematically continuous vectors of a fixed dimension (called embeddings), NE makes it possible to apply advanced machine learning algorithms to network data (Perozzi et al., 2014). Numerous studies have shown that by encoding semantic features (e.g. neighborhood information, topology similarity, etc.) into node embeddings, machine learning (ML) approaches can significantly outperform conventional non-ML methods on various downstream tasks in network analysis including node classification (Tang et al., 2015; Perozzi et al., 2014), link prediction (Wang et al., 2016), network alignment (Liu et al., 2016; Man et al., 2016), community detection (Keikha et al., 2018), etc. Different from the current popular graph neural networks that require node attributes as the input (Kipf & Welling, 2016; Hamilton et al., 2017), NE plays an unsubstitutable role in dealing with non-attributed networks.

Low-dimensional embedding is a natural idea for NE which starts from the perspective of dimension reduction (Belkin & Niyogi, 2002; Roweis & Saul, 2000; Yan et al., 2006). It has been well studied academically in the decades (Tang et al., 2015; Perozzi et al., 2014; Grover & Leskovec, 2016), and widely adopted in industry due to the good efficiency and scalability (Zhang et al., 2018a). Although low-dimensional (e.g. 128-dim) NE has become a basic principle in literature (Goyal & Ferrara, 2018), there is the problem that low-dimension embedding is merely a low-rank approximation of some information matrix describing how nodes interact that is usually high-rank. Some works

[*]Correspondance author. This work was in part supported by NSFC (62222607, 72342023) and SJTU Trans-med Awards Research (STAR) 20210106.

(Goyal & Ferrara, 2018; Yin & Shen, 2018) tried increasing the embedding directly but ended with a common finding that high-dimensional embeddings suffer from the problem of over-fitting and the subsequent descent of model performance in the inference tasks. Another drawback of a high embedding dimension is that the high computational complexity scales linearly to the dimension, which afterward would cause problems such as increasing training time and reducing inference efficiency (Yin & Shen, 2018; Wu et al., 2016). In this paper, we are going to overturn these common beliefs by presenting a brand new NE paradigm which is high-dimensional but endowed with a low computational complexity (Sec. 3), competent in the preservation of high-rank information (Sec. 5.2), showing good performance on both information preservation and inference (Sec. 6).

Embedding in the exponentially large quantum Hilbert space was firstly proposed in word2ket[1] (Panahi et al., 2020) for model quantization. By representing the word embeddings as 'pseudo' entangled states (see Appendix E.2 for interpretation of 'pseudo' entangled quantum state) whose dimension grows exponentially as the amount of used parameters, it achieves to save parameters in the embedding model. Since the adopted neural decoder suffers a high complexity for high-dimensional embeddings, word2ket is restricted in a low-dimensional embedding space where the dimension is up to 8,000 as reported in experiments (much smaller than the vocabulary size). Though high-dimensional embedding is not explored in word2ket (Panahi et al., 2020), it sheds light on a new representation form that uses fewer parameters than the dimension to construct embeddings.

Compared with entangled states, product states can generate the same high-dimensional embedding with fewer parameters, with the complexity linear to the number of parameters in computing the fidelity (equivalent to inner product). Therefore, in the proposed high-dimensional NE paradigm **node2ket**, we represent node embeddings as the product states (Sec. 3) and adopt inner product-based objectives for embedding training to achieve a low complexity. We also find that extra constraints are necessary since the tensorized embedding brings instability to model learning. These constraints include that all the embeddings are normalized (for stability) and the inner product between embeddings should be positive (for interpretability, see Sec. 4.1 for more details). Due to these constraints, the origin SGD which empirically works well for previous conventional NE methods (Tang et al., 2015; Perozzi et al., 2014; Grover & Leskovec, 2016) would fail in embedding optimization of node2ket. To make the optimization procedure work and improve the robustness of the embedding learning process under different possible application scenarios while balancing the running efficiency, we develop an optimizer for stable learning of high-dimensional NE, which learns embeddings on the unit hypersphere by Riemannian optimization, keeps monitoring inner products between embeddings and makes them positive, and updates embeddings by Adagrad in asynchronous batches (Sec. 4). Based on node2ket, we propose a variant node2ket+ which introduces parameter sharing in the embedding construction step of node2ket (Sec. 3.1). Compared with existing works of NE, node2ket+ can explicitly encode semantic information and is endowed with a significantly higher time/space efficiency. **The main contributions are:**

1) At the algorithm level, we open the path to efficient high-dimensional embedding which has been rarely studied in NE literature before, and specifically devise two NE methods **node2ket** and **node2ket+** as a starting point. We theoretically analyze the essence of our methods from the perspective of implicit matrix factorization and give a sufficient condition when it produces a full-rank approximation of the latent information matrix of the given network.

2) At the engineering level, we implement and integrate all the proposed techniques into a C++ library named as ***LIBN2K***, which is a fundamental toolkit for efficient high-dimensional NE algorithm development with the following features as a robust tool for future research in NE:

- **Flexible and robust.** It is highly flexible with plug-in modules including flexible embedding construction, sampling strategies, objectives, two types of embedding architectures. The developed optimizer designed to deal with the proposed two necessary constraints ensures that the system runs stably and robustly in different scenarios.
- **Highly efficient and scalable.** It runs in a distributed manner with multiple CPU threads, which enables it to handle large-scale networks on a single machine. The only-CPU style makes it easy to be deployed on most devices where only CPUs are available.
- **Compatible with both networks and sequences.** Considering that most NE methods either learn from a highly-structured network or implicitly structured node sequences, it accepts both the two

---

[1] 'ket' is a representation of quantum states, see Appendix E.1 for more details.

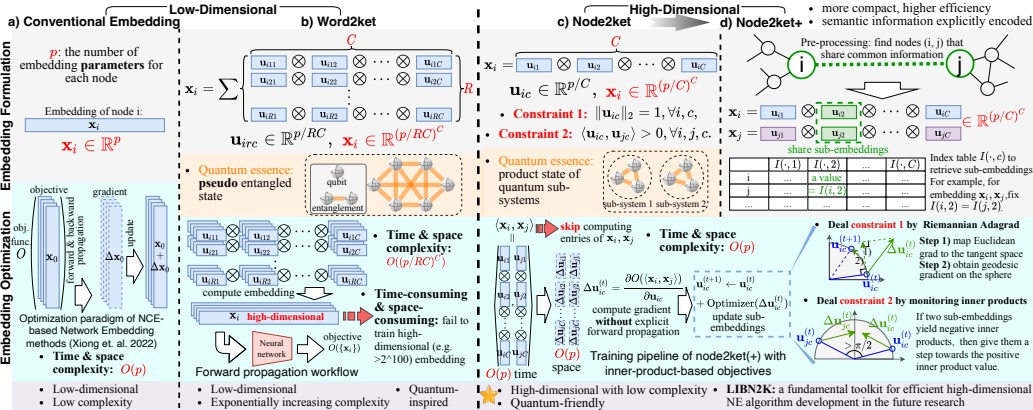

Figure 1: Technical points of node2ket compared with conventional embedding and word2ket.

types of data as inputs. Learning from node sequences in streams also further decreases memory usage since the whole network is not loaded into the memory.

3) Experiments are conducted on downstream tasks including network reconstruction, link prediction, and node classification. We show the overwhelming performance of our methods that use much fewer parameters to achieve the best scores among all the baselines. The high running speed, low memory usage, and low local space consumption for embedding storage especially for large-scale networks are impressive. Note our setting follows the widely adopted node attribute-free protocol which in fact can avoid privacy breaking, and thus GNNs are not compared as done in peer literature.

## 2 RETROSPECT OF EXISTING METHODS

We define a non-attributed network as $G = (\mathcal{V}, \mathcal{E})$ where $\mathcal{V} = \{i\}_{i=1}^N$ is the node set and $\mathcal{E} \subseteq \mathcal{V} \times \mathcal{V}$ is the edge set. NE aims to represent each node $i \in \mathcal{V}$ as a continuous vector. Suppose that for each node $i$ we use $p$ parameters to represent the embedding $\mathbf{x}_i$. Based on the above framework, different existing embedding methods can be described as follows.

**Conventional Embedding, Fig. 1 a).** Embedding $\mathbf{x}_i$ is directly constructed as a vector $\mathbf{x}_i \in \mathbb{R}^p$ with the $p$ parameters ($p \ll N$). Because of its simplicity and low time & space complexity that are linear to $p$ for model optimization, NE methods based upon the convention embedding are usually easy to be implemented, time-efficient and scalable. With different objectives and sampling strategies over diverse types of data and application scenarios, the conventional embedding is adopted as the default in all works in NE to the best of our knowledge (see Appendix B for related works).

**Word2ket, Fig. 1 b).** Divide the $p$ parameters into $R$ rows and $C$ columns, then each block on $r$-th row $c$-th column contains $p/RC$ parameters formulating a 'sub-embedding' $\mathbf{u}_{irc} \in \mathbb{R}^{p/RC}$. The node embedding is constructed by $\mathbf{x}_i = \sum_{r=1}^R \bigotimes_{c=1}^C \mathbf{u}_{irc} \in \mathbb{R}^{(p/RC)^C}$ where $\otimes$ denotes the tensor product (equal to outer product here). In physical view, the formulation of $\mathbf{x}_i$ looks like an entangled quantum state but not a true one (details in Appendix E.2). It is more precise to be interpreted as a rank-$R$ order-$C$ CP Decomposition in mathematics (Hitchcock, 1927). During the model optimization, a concrete $\mathbf{x}_i$ should be obtained as the input to the following neural network-based decoder. That causes two problems: **i) High time & space complexity:** an $O((p/RC)^C)$ time complexity for the decoder to process $\mathbf{x}_i$ and an $O((p/RC)^C)$ space complexity to store the temporal variable $\mathbf{x}_i$ in the memory; **ii) Low embedding dimension:** Due to the high complexity, the seemingly high embedding dimension should be low enough to afford. Furthermore, because $\mathbf{x}_i$ is not strictly a quantum state and neural networks are not quantum-friendly word2ket is still a quantum-inspired algorithm but not one that can be implemented by quantum computing.

## 3 EMBEDDING CONSTRUCTION BY NODE2KET

**Embedding Construction** (shown in Fig. 1 c). We divide the $p$ parameters (referred in Sec. 2) into $C$ columns, with each column containing $p/C$ parameters which defines a sub-embedding $\mathbf{u}_{ic} \in \mathbb{R}^{p/C}$. Then the embedding is constructed by $\mathbf{x}_i = \bigotimes_{c=1}^C \mathbf{u}_{ic} \in \mathbb{R}^{(p/C)^C}$. With the constraint

$\|\mathbf{u}_{ic}\|_2 = 1$ introduced in Sec. 4.1, embedding by node2ket will be a strict 'product state' following the normalization rule $\|\mathbf{x}_i\|_2 = 1$. The constraint distinguishes node2ket's embeddings from the embeddings of rank-1 version of word2ket.

**The $O(p)$ Time and Space Complexity in Training.** To achieve the $O(p)$ complexity in training, the objective function must be based on the inner product, which is a very loose condition and indeed very common in NE objectives, e.g. the Skip-Gram with Negative Sampling (SGNS) and the Marginal Triplets (MT) (see Appendix C.2 for details). The $O(p)$ complexity of node2ket comes from the convenience in inner product computation which is given by:

$$\langle \mathbf{x}_i, \mathbf{x}_j \rangle = \prod_{c=1}^{C} \langle \mathbf{u}_{ic}, \mathbf{u}_{jc} \rangle, \tag{1}$$

where each $\langle \mathbf{u}_{ic}, \mathbf{u}_{jc} \rangle$ takes $O(p/C)$ time. Since the embedding $\mathbf{x}_i$ and $\mathbf{x}_j$ are not computed, the space complexity is also $O(p)$. Furthermore, in other cases such as computing Euclidean distance, the time and space complexity are also $O(p)$ by $\|\mathbf{x}_i - \mathbf{x}_j\|_2 = \sqrt{\langle \mathbf{x}_i, \mathbf{x}_i \rangle + \langle \mathbf{x}_j, \mathbf{x}_j \rangle - 2\langle \mathbf{x}_i, \mathbf{x}_j \rangle}$.

### 3.1 *node2ket+*: MORE COMPACT WITH SEMANTIC INFORMATION EXPLICITLY ENCODED

Conventionally, semantic information (e.g. various orders of proximity (Cao et al., 2015), structural similarity (Ribeiro et al., 2017), etc.) are preserved by designing a fancy sampling strategy that **implicitly** defines nodes' similarities in the embedding space. As node2ket divides the parameters into sub-embeddings, it provides an opportunity to **explicitly** encode semantic information in the step of embedding construction, which is achieved by parameter sharing between nodes that share common information. As illustrated in Fig. 1 d), if we have found two semantically proximate nodes $i$ and $j$, then we can share the sub-embeddings between them. To technically achieve that, we need to assign each node $C$ indices to retrieve the sub-embeddings. We use a table $I(i, c)$ to record the index of $c$-th sub-embedding of node $i$. Then if $I(i, c) = I(j, c)$, embeddings $\mathbf{x}_i$ and $\mathbf{x}_i$ shares the same $c$-th sub-embedding. In the example of Fig. 1 d), we share the parameters of $\mathbf{u}_{i2}$ and $\mathbf{u}_{j2}$ by fixing $I(i, 2) = I(j, 2)$. The parameter sharing scheme of node2ket+ would also reduce space overhead. According to this point, we apply node2ket+ on compressive network embedding.

### 3.2 APPLY NODE2KET+ ON COMPRESSIVE NETWORK EMBEDDING

One promising applications of node2ket+ is compressive NE (CNE) especially for devices where only limited computational resources are available. By parameter sharing, the memory usage and local space for embedding storage can be significantly reduced. One feasible approach for CNE is to equip node2ket+ with graph partition algorithms, e.g. Louvain partition (Blondel et al., 2008), to generate $I(i, c)$ table. Louvain partition algorithm is a fast and scalable unsupervised partition algorithm that can divide a network into parts with a given resolution $r$. A higher resolution will result in a larger number of partitions. To explore community features of different granularity, we run multiple times of Louvain partition with different resolutions. If two nodes $i$ and $j$ are divided into a common community in the $c$-th time of Louvain partition, then we set $I(i, c) = I(j, c)$.

The following Proposition 1 shows the lower bound of the number of parameters to train a fixed $q$-dimensional NE. The proof is given in Appendix D.1. As a concrete example, when $N = 10^6$ and $q = 100$, by setting $C = 4$, we only need 400 parameters to store the sub-embeddings and $4 \times 10^6$ to store index table $I(i, c)$, far less than conventional embedding methods that require $10^8$ parameters.

**Proposition 1** (Capability of Node2ket+ on Embedding Compression). *To train $q$-dimensional embeddings for $N$ nodes, if it requires obtaining $N$ different embeddings $\{\mathbf{x}_i\}_{i=1}^{N}$, by node2ket+, we need at least $C(Nq)^{1/C}$ parameters to store sub-embeddings and $NC$ parameters to store the index table $I(i, c)$, where $C$ is the number of sub-embeddings for each node.*

## 4 LEARNING SUB-EMBEDDINGS UNDER CONSTRAINTS

In general, a typical NE algorithm can be formalized as iterations of the following steps (illustrated in Fig. 4, Appendix C): i) sample a node pair $(i, j)$ as the positive sample by some strategy $P(i, j|G)$ (e.g. directly sample from edges), ii) compute the gradients according to the objective $O(i, j)$ defined over $i$ and $j$, and iii) update node embeddings given gradients. Hence the complete objective of such an NE algorithm can be described as follows:

$$\text{maximize} \ \ \mathbb{E}_{(i,j) \sim P(i,j|G)} O(i, j). \tag{2}$$

Great efforts of research in NE have been paid in designing expressive objectives, and exploring network structures by diverse task-oriented sampling strategies (see Appendix B for related works). We do not pay incremental efforts on these two aspects, but aim to provide a general approach for model optimization that works for any *inner product-based* objectives. By 'inner product-based objective' we refer to an objective that is a function of inner products between one or multiple embedding pairs. In this paper, we investigate two sampling strategies, random walk (RW) and random walk with restart (RWR) (details in Appendix C.1), and two inner product-based objective functions, skip-gram with negative sampling (SGNS) and marginal triplets (MT) (details in Appendix C.2).

In the following part of this section, we will discuss two constraints in embedding learning, for which we will also show how to optimize the derived constrained objective.

### 4.1 CONSTRAINTS IN EMBEDDING LEARNING

**The Normalization Constraint** $\|\mathbf{u}_{ic}\|_2 = 1, \; \forall i, c$. With the constraint, the embedding can become a strict quantum state (see Appendix E.3), which makes the algorithm quantum-friendly since the embedding can be loaded into a quantum machine without further normalization and the inner product can be obtained by computing the fidelity of quantum states, e.g. by a simple swap test (Barenco et al., 1997) or quantum matrix multiplication (Shao, 2018). The constraint is also empirically necessary for stable embedding learning, preventing the sub-embeddings from collapsing to zero vectors that would make the program crash.

**The Positive Inner Product Constraint** $\langle \mathbf{u}_{ic}, \mathbf{u}_{jc} \rangle > 0, \; \forall i, j, c$. In Eq. 1, the inner product of two embeddings $\langle \mathbf{x}_i, \mathbf{x}_j \rangle$ can be decomposed into the product of the inner products of the sub-embeddings $\langle \mathbf{u}_{ic}, \mathbf{u}_{jc} \rangle$. So it is rather likely to happen that 'two negatives make a positive', e.g. $\langle \mathbf{u}_{i1}, \mathbf{u}_{j1} \rangle < 0, \langle \mathbf{u}_{i2}, \mathbf{u}_{j2} \rangle < 0$ yielding $\langle \mathbf{x}_i, \mathbf{x}_j \rangle = \langle \mathbf{u}_{i1}, \mathbf{u}_{j1} \rangle \langle \mathbf{u}_{i2}, \mathbf{u}_{j2} \rangle > 0$. Such cases bring chaos in model optimization and also make the embedding results less interpretable for visualization.

**The Objective Under Constraints.** The objective Eq. 2 under the above two constraints now is:

$$\text{maximize} \; \mathbb{E}_{(i,j) \sim P_{(i,j|G)}} O(i,j),$$
$$\text{s.t.} \; \|\mathbf{u}_{ic}\|_2 = 1, \forall i, c, \; \text{and} \; \langle \mathbf{u}_{ic}, \mathbf{u}_{jc} \rangle > 0, \forall i, j, c. \tag{3}$$

### 4.2 GRADIENT COMPUTING

Note that the gradient computing here is very different from the explicit forward/backward propagation as done in the deep learning frameworks such as Pytorch and Tensorflow. Specifically, we compute the gradients of sub-embedding $\mathbf{u}_{ic}$ directly through mathematical derivation. In this way, the step of computing the loss can be skipped thus accelerating training. And most importantly, the computation of $\mathbf{x}_i$ is skipped, which makes training embeddings in the super high-dimensional (can be even larger than $2^{128}$-dimensional) Hilbert space possible on a single machine.

For embedding $\mathbf{x}_i = \bigotimes_{c=1}^{C} \mathbf{u}_{ic}, \mathbf{x}_j = \bigotimes_{c=1}^{C} \mathbf{u}_{jc}$, the partial derivative of the inner product-based objective $O(i,j)$ w.r.t. the sub-embedding $\mathbf{u}_{ic}$ is:

$$\frac{\partial O(i,j)}{\partial \mathbf{u}_{ic}} = \frac{\partial O(i,j)}{\partial \langle \mathbf{x}_i, \mathbf{x}_j \rangle} \frac{\partial \langle \mathbf{x}_i, \mathbf{x}_j \rangle}{\partial \mathbf{u}_{ic}} = \frac{\partial O(i,j)}{\partial \langle \mathbf{x}_i, \mathbf{x}_j \rangle} \mathbf{u}_{jc} \prod_{c' \neq c} \langle \mathbf{u}_{ic'}, \mathbf{u}_{jc'} \rangle, \tag{4}$$

where $\frac{\partial O(i,j)}{\partial \langle \mathbf{x}_i, \mathbf{x}_j \rangle}$ depends on the objective (see Appendix C.2 for examples of MT and SGNS).

### 4.3 TRAINING PIPELINE

We optimize Eq. 3 by asynchronously updating embeddings on CPUs in line with the major literature on network embedding (Mikolov et al., 2013b; Tang et al., 2015; Perozzi et al., 2014; Grover & Leskovec, 2016). Although the embedding optimization by asynchronous SGD (ASGD) (Recht et al., 2011) empirically works well for conventional embedding methods, the simple ASGD fails in our method where the gradients are much more complex with further constraints. Thus we introduce new techniques to optimize the proposed compressive embedding model asynchronously with multiple CPU threads under the constraints stated in Sec. 4.1. Details of an iteration of embedding training is shown in Alg.1. The features and explanations of Alg. 1 are listed as follows:

**i) Embeddings are learned on the unit hypersphere** (see the function `RiemannAdagrad` in Alg. 1, for motivation please recall the normalization constraint in Sec. 4.1). For an embedding $\mathbf{u}$,

we denote its Euclidean gradient as $\texttt{Grad}(\mathbf{u})$. We first apply the Adagrad algorithm (Duchi et al., 2011) to compute the updating step $\Delta\mathbf{u}$ for embedding $\mathbf{u}$. Adagrad can automatically modify the step size of $\Delta\mathbf{u}$ and accelerate model convergence. Our target is to update $\mathbf{u}$ according to $\Delta\mathbf{u}$ with the constraint that the updated embedding still falls on the hypersphere. As shown in lines 3-8 of the function $\texttt{RiemannAdagrad}$ in Alg. 1, we introduce three optimization strategies to achieve the target. **In the order-0 strategy,** we update the embedding $\mathbf{u}$ in Euclidean space by the step $\Delta\mathbf{u}$ and then map the updated embedding back to the hypersphere; **in the order-1 strategy,** we first map $\Delta\mathbf{u}$ to the tangent space of $\mathbf{u}$ by $\widehat{\Delta\mathbf{u}} = \Delta\mathbf{u} - \langle\mathbf{u}, \Delta\mathbf{u}\rangle\mathbf{u}$, then update $\mathbf{u}$ by the step $\widehat{\Delta\mathbf{u}}$, and finally normalize the updated embedding; **in the order-2 strategy,** we adopts *exponential mapping* (Manfredo, 1992) which moves $\mathbf{u}$ in the direction of $\widehat{\Delta\mathbf{u}}$ with velocity $\|\widehat{\Delta\mathbf{u}}\|$ along the geodesic of the manifold. The number of computational steps increases with the order of the updating strategy.

**ii) Inner products between embeddings are mostly positive** (see line 3-9 of function $\texttt{node2ketIter}$ in Alg. 1, for motivation please recall the second constraint $\langle\mathbf{u}_{ic}, \mathbf{u}_{jc}\rangle > 0, \ \forall i, j, c$ in Sec. 4.1). We deal with this constraint by maximizing the term $\sum_{i,j\in\mathcal{V}}\sum_{c=1}^{C}\min(\langle\mathbf{u}_{ic}, \mathbf{u}_{jc}\rangle, 0)$. In practice, we keep monitoring the inner products between embeddings during the embedding learning procedure. If we find a pair of embedding $\langle\mathbf{u}_{ic}, \mathbf{u}_{jc}\rangle < 0$, we maximize their inner product.

**iii) Asynchronous batch optimization is adopted** (see the function $\texttt{BatchUpdate}$ in Alg. 1). Though adopted by previous works and working well in conventional embedding methods, updating embeddings once after obtaining gradients (Tsitsulin et al., 2018; Perozzi et al., 2014; Grover & Leskovec, 2016) can be too frequent for embedding learning thus causing a great overhead during training. On the other hand, gradients of an embedding can be very dissimilar for different positively or negatively paired nodes, which brings difficulties to model convergence. We asynchronously implement the batch optimization algorithm with a variable $\texttt{Count}(\mathbf{u})$ recording how many times that 'back-propagation' is performed on the embedding $\mathbf{u}$. When $\texttt{Count}(\mathbf{u})$ reaches the batch size, the program updates the embedding $\mathbf{u}$.

---

**Algorithm 1:** An iteration of node2ket.

---

**Input**: sub-embedding dimension $d$, number of sub-embeddings for each node $C$, sampling function $P(i|G)$ and $P(j|i, G)$, initial learning rate $\eta = 0.2$, batch size $bs = 8$;

**function** $\texttt{node2ketIter}$

1: Sample a node pair $(i, j)$ as the positive sample by $i \sim P(i|G)$, $j \sim P_{pos}(j|i, G)$;
2: Sample $K$ nodes $\{j'|j' \sim P_{neg}(j'|G)\}$ so that $(i, j')$ form negative samples (positive/negative samples are concepts from constrastive learning Chen et al. (2020));
3: ▶ Positive inner product constraint
4: **for** $n \in \{j\} \cup \{j'\}^K, c = 1, \cdots C$ **do**
5:     **if** $\langle\mathbf{u}_{ic}, \mathbf{u}_{nc}\rangle < 0$ **then**
6:         $\texttt{Grad}(\mathbf{u}_{ic}) \leftarrow \texttt{Grad}(\mathbf{u}_{ic}) + \frac{\partial\langle\mathbf{u}_{ic}, \mathbf{u}_{nc}\rangle}{\mathbf{u}_{ic}}$;
7:         $\texttt{Grad}(\mathbf{u}_{nc}) \leftarrow \texttt{Grad}(\mathbf{u}_{nc}) + \frac{\partial\langle\mathbf{u}_{ic}, \mathbf{u}_{nc}\rangle}{\mathbf{u}_{nc}}$;
8:     **end if**
9: **end for**
10: ▶ Compute gradients and update embeddings
11: **for** $n \in \{i, j\} \cup \{j'\}^K, c = 1, \cdots, C$ **do**
12:     $\texttt{Grad}(\mathbf{u}_{nc}) \leftarrow \texttt{Grad}(\mathbf{u}_{nc}) + \frac{\partial O(i, j)}{\partial\mathbf{u}_{nc}}$;
13:     $\texttt{BatchUpdate}(\mathbf{u}_{nc}, \texttt{Grad}(\mathbf{u}_{nc}))$;   ▶ Update as a batch
14: **end for**

**function** $\texttt{BatchUpdate}(\mathbf{u}, \texttt{Grad}(\mathbf{u}))$
1: $\texttt{Count}(\mathbf{u}) \leftarrow \texttt{Count}(\mathbf{u}) + 1$;     ▶ Batch counter
2: **if** $\texttt{Count}(\mathbf{u}) \geq bs$ **then**
3:     $\texttt{Grad}(\mathbf{u}) \leftarrow \texttt{Grad}(\mathbf{u})/\texttt{Count}(\mathbf{u})$;
4:     $\texttt{RiemannAdagrad}(\mathbf{u}, \texttt{Grad}(\mathbf{u}))$  ▶ Normalization constraint
5:     $\texttt{Count}(\mathbf{u}) \leftarrow 0$;
6:     $\texttt{Grad}(\mathbf{u}) \leftarrow \mathbf{0}$;
7: **end if**

**function** $\texttt{RiemannAdagrad}(\mathbf{u}, \texttt{Grad}(\mathbf{u}))$
1: $\texttt{StateSum}(\mathbf{u}) \leftarrow \texttt{StateSum}(\mathbf{u}) + \|\texttt{Grad}(\mathbf{u})\|/d$;  ▶ Adagrad
2: $\Delta\mathbf{u} \leftarrow \frac{\eta}{\sqrt{\texttt{StateSum}(\mathbf{u})} + \epsilon}\texttt{Grad}(\mathbf{u})$     ▶ $\epsilon = 10^{-9}$
3: ▶ Learning embedding on the Riemann manifold
4: [Order=0] $\mathbf{u} \leftarrow \texttt{Norm}(\mathbf{u} + \Delta\mathbf{u})$;
5: ▶ Map $\Delta\mathbf{u}$ onto the tangent space of $\mathbf{u}$
6: [Order=1 or 2] $\widehat{\Delta\mathbf{u}} \leftarrow \Delta\mathbf{u} - \langle\mathbf{u}, \Delta\mathbf{u}\rangle\mathbf{u}$;
7: [Order=1] $\mathbf{u} \leftarrow \texttt{Norm}(\mathbf{u} + \widehat{\Delta\mathbf{u}})$;
8: [Order=2] $\mathbf{u} \leftarrow \cos(\|\widehat{\Delta\mathbf{u}}\|)\mathbf{u} + \frac{\sin(\|\widehat{\Delta\mathbf{u}}\|)}{\|\widehat{\Delta\mathbf{u}}\|}\widehat{\Delta\mathbf{u}}$;

---

## 5 THEORETICAL ANALYSIS

### 5.1 PRELIMINARIES: NETWORK EMBEDDING IS IMPLICIT MATRIX FACTORIZATION

Previous studies (Levy & Goldberg, 2014; Qiu et al., 2018) have shown that the essence of network embedding is implicit matrix factorization (IMF) over an information matrix $\mathbf{A}(G) \in \mathbb{R}^{N\times N}$ of the given network $G$. $\mathbf{A}(G)$ indicates the latent node proximity and is implicitly defined by the whole training procedure, including the sampling strategy, the objective, and also constraints over embeddings. There are two types of IMF for NE (Armandpour et al., 2019; Xiong et al., 2022):

$$\text{Type-I IMF: } \underset{\mathbf{X}}{\text{minimize}} \ \|\mathbf{A}(G) - \mathbf{X}^\top\mathbf{X}\|_F, \quad \text{Type-II IMF: } \underset{\mathbf{X},\mathbf{H}}{\text{minimize}} \ \|\mathbf{A}(G) - \mathbf{X}^\top\mathbf{H}\|_F, \quad (5)$$

where $\mathbf{X} = [\mathbf{x}_1, \mathbf{x}_2, \ldots, \mathbf{x}_N]$ is the node embedding matrix, and $\mathbf{H} = [\mathbf{h}_1, \mathbf{h}_2, \ldots, \mathbf{h}_N]$ is called the hidden embedding matrix (a.k.a. context embedding in some works (Mikolov et al., 2013b; Tang

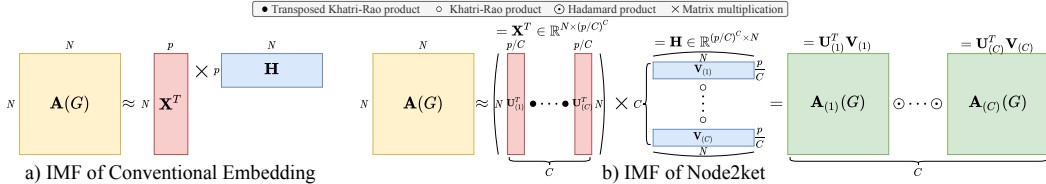

Figure 2: Implicit matrix factorization (IMF) of conventional embedding methods and node2ket.

et al., 2015)) which is constructed in the same way of $\mathbf{X}$. The type-II IMF achieves to encode asymmetric information while the type-I IMF can only preserve symmetric information. Some works also that the type-I objective focuses on the first-order information of $\mathbf{A}(G)$ while type-II emphasizes more on the second-order information (Tang et al., 2015; Xiong et al., 2022). Technically, the type-I IMF can be regarded as a special case of the type-II IMF by fixing $\mathbf{H} \equiv \mathbf{X}$.

## 5.2 NODE2KET PERFORMS HIGH-RANK APPROXIMATION OF THE INFORMATION MATRIX

Since the type-II IMF is more general, we take the type-II IMF as the example for analysis while the conclusion also fits the type-I IMF.

**Conventional Embedding is a Low-Rank Approximation of $\mathbf{A}(G)$ (illustrated in Fig. 2 a).** As shown in Sec. 2, in convention embedding the embedding matrix $\mathbf{X}$ and $\mathbf{H}$ have the shape $p \times N$, $p$ denoting the number parameters for each node. It is obvious that $rank(\mathbf{X}^\top \mathbf{H}) \leq p$. In real-world networks, $\mathbf{A}(G)$ can usually be very complex and high-rank. When $p \ll rank(\mathbf{A}(G))$ and the singular values of $\mathbf{A}(G)$ are close, only a very small portion of information can be preserved.

**Node2ket Performs High-Rank Approximation of $\mathbf{A}(G)$ (illustrated in Fig. 2 b).** We denote the $c$-th column sub-embedding matrix as $\mathbf{U}_{(c)} = [\mathbf{u}_{1c}, \mathbf{u}_{2c}, \ldots, \mathbf{u}_{Nc}]$ and the $c$-th column sub-hidden-embedding matrix as $\mathbf{V}_{(c)} = [\mathbf{v}_{1c}, \mathbf{v}_{2c}, \ldots, \mathbf{v}_{Nc}]$. Then by $\mathbf{x}_i = \bigotimes_{c=1}^{C} \mathbf{u}_{ic}$ and $\mathbf{h}_i = \bigotimes_{c=1}^{C} \mathbf{v}_{ic}$, we have $\mathbf{X} = \mathbf{U}_{(1)} \circ \cdots \circ \mathbf{U}_{(C)}$ and $\mathbf{H} = \mathbf{V}_{(1)} \circ \cdots \circ \mathbf{V}_{(C)}$ where $\circ$ denotes the Khatri-Rao product (a.k.a. column-wise Kronecker product) (Liu, 1999). Then we have the following formula:

$$\mathbf{X}^\top \mathbf{H} = \mathbf{A}_{(1)}(G) \odot \cdots \odot \mathbf{A}_{(C)}(G), \tag{6}$$

where $\odot$ denotes Hadamard product, $\mathbf{A}_{(c)}(G) = \mathbf{U}_{(c)}^\top \mathbf{V}_{(c)}$ is the information matrix reconstructed by $c$-th column of sub-embedding and sub-hidden-embedding with $rank(\mathbf{A}_{(c)}(G)) \leq p/C$. One of the properties of Hadamard product is that the rank can be exponentially increasing. More precisely, we have the following theorem over the rank of $\mathbf{X}^\top \mathbf{H}$ whose proof is put in Appendix D.2:

**Theorem 1.** *By node2ket, we denote* $\overline{\mathbf{A}}_{(c)}(G) = \bigodot_{k \neq c} \mathbf{A}_{(k)}(G)$, *we have* $rank(\mathbf{X}^\top \mathbf{H}) = N$ *if there* $\exists c$ *s.t.* $\mathbf{A}_{(c)}(G)$ *and* $\overline{\mathbf{A}}_{(c)}(G)$ *are positive semidefinite, and* $\max \left( K\text{-}rank(\mathbf{A}_{(c)}(G)) + rank(\overline{\mathbf{A}}_{(c)}(G)), rank(\mathbf{A}_{(c)}(G)) + K\text{-}rank(\overline{\mathbf{A}}_{(c)}(G)) \right) > N$, *and K-rank denotes the Kruskal rank.*

To intuitively see the supremacy of node2ket over conventional embedding in approximating high-rank information matrices given the same $p$, we give toy examples for both type-I (Example 1, Appendix D.3) and type-II (Example 2, Appendix D.4) IMF. Example 1 gives a $G$ of 36 nodes with an information matrix $rank(\mathbf{A}(G)) = 16$, for which conventional embedding gives a rank-8 approximation given $p = 8$. And Example 2 gives a $G$ of 9 nodes with an information matrix $rank(\mathbf{A}(G)) = 9$, for which conventional embedding gives a rank-6 approximation give $p = 6$. In comparison, node2ket can fully recover $\mathbf{A}(G)$ with the same $p$.

**Difficulties of Solving Node2ket by Explicit Matrix Factorization (EMF).** When $\mathbf{A}(G)$ can be explicitly expressed (e.g. (Cao et al., 2015; Qiu et al., 2018; 2019)), obtaining embeddings of node2ket by solving matrix factorization is still very difficult. Different from EMF by conventional embedding whose solution can be obtained through SVD for $\mathbf{A}(G)$, EMF by node2ket involves an NP-hard problem, rank-1 CP Tensor Decomposition (Hitchcock, 1927; Kiers, 2000). A feasible solution is: i) conduct SVD $\mathbf{A}(G) = \mathbf{P\Sigma Q}^\top$; ii) Expand $\mathbf{P}$ from the shape $N \times N$ to shape $N \times (p/C)^C$ by concatenating a zero matrix then do a transpose: $\mathbf{X} = [\mathbf{P}, \mathbf{0}]^\top$; iii) Conduct rank-1 CP Decomposition for each column of $\mathbf{X}$ which can be written as the Khatri-Rao product:

$\mathbf{X} = \mathbf{U}_{(1)} \circ \cdots \circ \mathbf{U}_{(C)}$. Step iii) is NP-hard (Phan et al., 2019; Krijnen et al., 2008), indicating the difficulties of node2ket by EMF.

## 6 EXPERIMENTS

### 6.1 PROTOCOL AND OVERVIEW

Details of settings including datasets, baselines, tasks, and metrics are in Appendix F.1.

**Hardware Configurations.** All of the experiments are run on a single machine with Intel(R) Core(TM) i9-7920X CPU @ 2.90GHz with 24 logical cores and 128GB memory.

**Datasets.** We study five public real-world datasets of different scales, different densities, and containing different types of information. All of the networks are processed as undirected ones. The detailed statistics of the networks are given in Table 5. We give a brief introduction of the datasets as follows: **YouTube** (Tang & Liu, 2009b) is a social network with millions of users on youtube.com; **BlogCatalog** (Tang & Liu, 2009a) (BC) is a social network of bloggers who have social connections with each other; **PPI** (Breitkreutz et al., 2007) is a subgraph of the PPI network for Homo Sapiens; **Arxiv GR-QC** (Leskovec et al., 2007) is a collaboration network from arXiv and covers scientific collaborations between authors with papers submitted to General Relativity and Quantum Cosmology category; **DBLP** (Ley, 2002) is a citation network of DBLP, where each node denotes a publication, and each edge represents a citation.

**Baselines.** The investigated NE methods include three based on training with SGNS objectives (LINE, node2vec, VERSE), two based on matrix factorization (ProNE and NetSMF), one based on hyperbolic embedding (HHNE), one based on graph partition (LouvainNE), one based on graph neural networks (GraphSAGE), and word2ket that constructs embeddings in the Hilbert space.

**Tasks.** We study three downstream tasks, including a preservation task, network reconstruction (NR), and two inference task, link prediction (LP), and node classification (NC).

We summarize all the experimental points and conclusions as follows:

- **Strong model performance against conventional baselines.** In Sec. F.2, we compare the performance of node2ket and node2ket+ with baselines with the same or varying number of parameters. Results on small networks demonstrate that our methods can achieve the best performance on all three tasks with the least parameters. We also find that as the dimension goes exponentially high, the model performance on all the three tasks NR, LP, and NC improves.

- In Sec. F.3, the **scalability** of the model in shown by experiments on a Million-scale network. Our methods achieve top-level performance, with impressive high running speed, low memory usage, and small space for embedding storage.

- **Performance with varying $C$ and $p$.** In Appendix F.4.1, we show that when either $C$ is too small ($C = 1$) or large ($C = p/2$) the performance degenerates, indicating that our embedding constitution as a product state of entangled states simultaneously improves the expressiveness and compactness.

- **Necessity of the proposed two constraints** is shown in Sec. F.4.2.

- **Fast convergence** brought by the Adagrad-based optimizer is demonstrated in Sec. F.4.3.

- **Robustness.** We demonstrate that our methods are robust with different sampling strategies (Appendix F.5.1) objectives (Appendix F.5.2), and types of embeddings (Appendix F.5.3).

- **Parallelizability.** In Sec. F.6 we show that the parallelization can decrease the running time significantly and will not affect model performance.

- **Visualization.** In Sec. F.7, we visualize the reconstructed adjacency matrix $\mathbf{A}_{(c)}(G)$, which shows that sub-embeddings $\mathbf{u}_{ic}$ with different index $c$ can orthogonally preserve the latent information matrix, which makes the approximated $\mathbf{A}(G)$ a high-rank one.

We tend to conclude that the excellent performance of node2ket comes from two aspects:

- Using the same amount of parameters, the embedding dimension of node2ket in the Hilbert space is much higher than conventional embedding methods, resulting in stronger expressiveness. Our implementation also carefully avoids the computation of high-dimensional embeddings.

Table 1: Network reconstruction precisions on medium-scale networks. 'CR.' is short for compressive ratio (see main text for the detailed definition).

|  | BC | DBLP | GR-QC | PPI | YTC |
|---|---|---|---|---|---|
| LINE | 4.80 | 2.34 | 54.19 | 15.18 | 0.10 |
| node2vec | 7.05 | 18.74 | 59.92 | 43.46 | 1.00 |
| VERSE | 10.96 | 16.96 | 56.08 | 27.59 | 7.26 |
| NetSMF | 7.64 | 18.08 | 30.26 | 17.14 | 0.28 |
| ProNE | 10.10 | 17.76 | 56.37 | 24.02 | 0.15 |
| LouvainNE | 6.79 | 21.33 | 43.47 | 17.77 | 11.90 |
| HHNE | 0.69 | 11.40 | 50.50 | 18.23 | <0.01 |
| GraphSAGE | 0.88 | 0.23 | 0.93 | 0.91 | 0.25 |
| w2k+n2k | 29.77 | 51.25 | 80.98 | 35.77 | 26.37 |
| node2ket | 50.83 | 88.97 | 97.90 | 81.56 | 74.62 |
| node2ket+ | **54.61** | **89.63** | **98.07** | **87.35** | **76.43** |
| CR. | 0.86 | 0.72 | 0.76 | 0.88 | 0.74 |

Table 2: Link prediction precisions on medium-scale networks. 'CR.' is short for the compressive ratio (see main text for the detailed definition).

|  | BC | DBLP | GR-QC | PPI | YTC |
|---|---|---|---|---|---|
| LINE | 53.08 | 75.45 | 95.86 | 55.15 | 62.76 |
| node2vec | 77.58 | 82.09 | 91.72 | 77.58 | 76.78 |
| VERSE | 33.35 | 68.81 | 93.79 | 51.55 | 59.83 |
| NetSMF | 59.34 | 82.09 | 87.59 | 72.16 | 74.58 |
| ProNE | 75.27 | 85.31 | 93.79 | 73.97 | 81.07 |
| LouvainNE | 57.99 | 73.44 | 86.90 | 57.99 | 82.11 |
| HHNE | 38.77 | 85.92 | 93.10 | 65.72 | 83.89 |
| GraphSAGE | 51.5 | 50.91 | 60.69 | 47.68 | <0.01 |
| w2k+n2k | 86.57 | 94.37 | 95.86 | 84.79 | 90.59 |
| node2ket | 89.28 | **95.37** | **96.55** | **84.54** | **91.63** |
| node2ket+ | **89.31** | 94.97 | 93.79 | **84.54** | **91.63** |
| CR. | 0.83 | 0.69 | 0.73 | 0.84 | 0.70 |

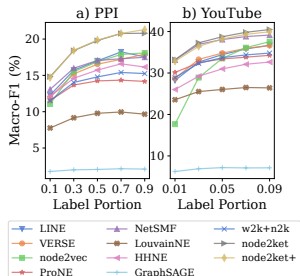

Figure 3: Results of node classification. The compressive ratio of node2ket+ is 0.90 on PPI, and 0.73 on YouTube.

- The sampling strategy and objective function design are important. For the sampling strategy, experiments show that the model prefers low-order information of the network (e.g. with a smaller window size by random walk as shown in Sec. F.5.1). For the objective, experiments in Sec. F.5.2 show that the MT loss is more suitable for structural tasks NR and LP, while the logistic loss is more suitable for the semantic task NC.

## 6.2 BRIEF EXPERIMENTAL RESULTS ON SMALL-SCALE NETWORKS

We evaluate all the baselines and node2ket on the three tasks, NR, LP, and NC. In NR and LP, we set $p = 128$ for all methods, $C = 4$ and $R = 2$ for w2k+n2k, and $C = 8$ for our methods. In the experiments of NC, we set $p = 32$ for all methods, $C = 2$ and $R = 2$ for w2k+n2k, and $C = 4$ for our methods. For node2ket+, we conduct Louvain partition on the former 4 sub-embeddings with resolution [100, 500, 1000, 1500], and do not do partition on the later 4 sub-embeddings. The compressive ratio of node2ket+ in Table 1 and Table 2 is defined as the ratio of the number of parameters used by node2ket+ to that used by other methods.

**Analysis for Results.** We give the results of NR in Table 1, LP in Table 2, NC in Fig. 3. **i)** Our methods consistently outperform the baselines on all three tasks, demonstrating the superiority of our embedding construction in the high-dimensional Hilbert space compared with other low-dimensional NE methods with the same amount of parameters. **ii)** By comparing node2ket and node2ket+, the compressive node2ket+ outperforms in NR, comparable in NC and LP. This indicates that the utilization of community information brings benefits to the preservation task NR and that appropriate compression won't have an adverse influence. **iii)** The embeddings learned by the GNN model GraphSAGE do not perform well on the three tasks. It is because GNN models highly rely on node features, which makes it difficult applying GNN methods on non-attributed networks.

## 7 CONCLUSIONS AND FUTURE WORKS

This paper makes contributions in two folds: On the algorithm level, we formalize the problem of high-dimensional network embedding, propose two approaches **node2ket** and **node2ket+** to constitute embeddings, and develop an optimizer to learn the embeddings stably with introduced constraints in embedding learning; On the engineering level, we develop a fundamental toolkit in c++ named as **LIBN2K** for further high-dimensional NE algorithm development.

There are also limitations for future work. **On geometry space of embeddings.** Though node2ket and node2ket+ are designed in quantum Hilbert space that is a high-dimensional spherical space, the approach of constructing compressive embeddings via tensor product is also applicable in other geometric spaces (e.g. Euclidean space, hyperbolic space). However, it may require specific model optimization techniques for embedding learning. This paper only discusses optimization in the quantum Hilbert space. **On exploitation of graph properties.** Previous works (see Sec. B) have developed numerous methods exploiting graph structures by designing novel sampling techniques. We only show that our methods work robustly in different cases, e.g. random walk and random walk with restart, but do not pay extra attention to structure exploitation. **On dealing with attributed networks.** Just as many NE methods, the methods are not designed for attributed networks.

## REPRODUCIBILITY STATEMENTS

The used data, the source code of node2ket, node2ket+, and the LIBN2K library, the compilation instructions, and all the scripts to run experiments in Sec. 6, are provided in the link shown right below author names in the first page.

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

# Appendix

## CONTENTS

# A  NOTATIONS

The used notations are listed as Table 3.

Table 3: Main notations and descriptions.

| Notations | Descriptions |
|---|---|
| $G = (\mathcal{V}, \mathcal{E})$ | Non-attributed network $G$ with the vertex set $\mathcal{V}$ and the edge set $\mathcal{E}$ |
| $\lvert \psi \rangle$ | Quantum states |
| $p$ | The number of parameters to construct an embedding |
| $q$ | The embedding dimension (when fixed in Sec. 3.2) |
| $\mathbf{x}_i$ | Embedding of node $i$ |
| $\mathbf{u}_{ic}$ | $c$-th sub-embedding of node $i$ |
| $N$ | Number of nodes of a network $G$ |
| $C$ | Number of sub-embeddings of each node embedding |
| $d$ | Dimension of $\mathbf{u}_{ic}$ ($d = p/C$ in node2ket) |
| $I(i,c)$ | (for node2ket+) the index of $\mathbf{u}_{ic}$ in the look-up table |
| $O(i,j)$ | Objective of a positive node pair $(i,j)$ |
| $\gamma$ | The 'margin' for the marginal triplets objective |
| $K$ | Number of negative samples for the skip-gram with negative sampling objective |
| $P(i,j\lvert G)$ | Probability that $(i,j)$ is sampled as a positive sample |
| $w$ | Window size for random walk |
| $\alpha$ | Probability that random walk restarts |
| $\Delta \mathbf{u}$ | The updating step of $\mathbf{u}$ in the Euclidean space |
| $\widehat{\Delta \mathbf{u}}$ | The updating step of $\mathbf{u}$ in the tangent space |

# B  RELATED WORKS

**Network Embedding.** Previous works of NE can be divided into works on algorithm design and works on efficient NE systems. **Algorithms.** NE methods based on the language model word2vec (Mikolov et al., 2013b) are one of the biggest branches of the NE research. In this branch, NE methods have been designed for different types of networks, including homogeneous networks (Perozzi et al., 2014; Grover & Leskovec, 2016; Tang et al., 2015), heterogeneous networks (Dong et al., 2017) and multiplex networks (Liu et al., 2017; Qu et al., 2017; Xiong et al., 2021). By designing the input node sequences, NE methods achieve to preserve specific types of information, e.g. low-order proximity (LINE (Tang et al., 2015),) structure similarity (struc2vec (Ribeiro et al., 2017)), versatile similarity measures (VERSE (Tsitsulin et al., 2018)), and cross-network alignment relations (CENALP (Du et al., 2022)). Another line of NE research is developing NE methods based on matrix factorization. GraRep (Cao et al., 2015) solves the embedding by matrix factorization for random walk and Skip-Gram while also taking high-order proximity information into consideration. NetMF (Qiu et al., 2018) proposes to unify some word2vec-based NE methods including LINE, DeepWalk, and node2vec, etc. within a matrix factorization framework, and NetSMF (Qiu et al., 2019) treats the network embedding task as sparse matrix factorization. AROPE and HOPE (Zhang et al., 2018b; Ou et al., 2016) propose to preserve multi-order proximity by matrix factorization with some mathematical tricks. Methods based on deep neural networks (Wang et al., 2016) especially graph neural networks (Hamilton et al., 2017) are also influential in literature of NE.

**Systems and Toolkits.** Developing efficient network embedding systems is also important for industrial applications of NE algorithms. GraphVite (Zhu et al., 2019) uses GPUs to accelerate embedding training for algorithms including LINE, DeepWalk, and node2vec. LightNE (Qiu et al., 2021) achieves to train embeddings for billion-scale networks based on ProNE (Zhang et al., 2019) and NetSMF (Qiu et al., 2019) on a single machine of 1.5TB memeory. Though endowed with high efficiency, these works are highly packaged for specific NE algorithms and hardly can they serve as a basic tool for developing new NE methods. There are also some works that aim to provide tools for developing NE methods, e.g. Gensim (Řehůřek & Sojka, 2010), and DGL (Wang et al., 2019). Gensim is a library for word embedding training, based on which many famous NE methods including DeepWalk (Perozzi et al., 2014), node2vec (Grover & Leskovec, 2016), and struc2vec (Ribeiro et al., 2017) are developed. These works obtain node embeddings by feeding the carefully designed

Table 4: Comparison between the word embedding library Gensim (Řehůřek & Sojka, 2010) and our high-dimensional NE library LIBN2K.

|  | Gensim | Proposed **LIBN2K** |
|---|---|---|
| Field | Word embedding | Network embedding |
| Dimension | Low | High |
| Supported Objectives | Skip-Gram, CBOW | Skip-gram with negative sampling, marginal triplets |
| Optimizer | SGD | Constrained Riemannian Adagrad |
| Supported Data | Sequences | Networks and sequences |
| Sampler | N/A | Random walk, random walk with restart |
| Implemented Methods | DeepWalk (Perozzi et al., 2014), node2vec (Grover & Leskovec, 2016), struc2vec (Ribeiro et al., 2017), etc. | node2ket, node2ket+, w2k+n2k |
| Language | Cython | C++ |
| Common Points | Good scalability, high efficiency, distributed training, CPU-only | |

node sequences into gensim.word2vec. DGL is a library for general deep graph learning mostly on GNN models. DGL also provides implementation for NE methods including DeepWalk and LINE, with GPUs accelerating training.

In the aspects of algorithm design and system developing, we design the first high-dimensional NE algorithms node2ket(+) and develop a toolkit LIBN2K for future high-dimensional NE algorithm development. The significance of LIBN2K for high-dimensional NE is similar to Gensim for NE, where the technical differences between them are summarized in Table 4.

**Tensorized Models for Model Compressing.** Tensors are a generalization of vectors and matrices. Thanks to the capability of representing and manipulating high-dimensional subjects, tensor has been successfully applied to exploit complex physical systems (Nielsen & Chuang, 2002). Noticing that tensors can be decomposed thus reducing the total number of parameters of models, researchers start to explore the application of tensor decomposition techniques in model compression. These works include compressing word embeddings in NLP (Panahi et al., 2020; Hrinchuk et al., 2020; Gan et al., 2022), compressing transformer models (Ma et al., 2019), learning high-order features by Graph Neural Networks (Hua et al., 2022). Compared with word2ket (Panahi et al., 2020) which adopts a similar embedding construction approach for deep neural networks, we substantially decrease computational overhead especially memory usage by skipping the computation of full-dimensional embeddings, thus fully utilizing the strong expressiveness of super-high dimensional Hilbert space (our $p > 2^{32}$ v.s. $p = 8000$ in word2ket). The tensorized models inspire us to conduct high-dimensional NE with a small number of parameters.

## C INVESTIGATED SAMPLING STRATEGIES AND LEARNING OBJECTIVES

### C.1 SAMPLING STRATEGIES

The sampling strategies $P(i, j|G)$ directly affect the performance of NE by implicitly defining proximity between the nodes. We decompose the probability as $P(i, j|G) = P(i|G)P(j|i, G)$, where $P(i|G) = \frac{\deg(i)}{\sum_{i'} \deg(i')}$ is the probability node $i$ being sampled as the first node with $\deg(i)$ referring to the degree of node $i$, and $P(j|i, G)$ is the probability that node $j$ is sampled as a positive node for $i$. We consider two strategies of $P(j|i, G)$, random walk (RW) and random walk with restart (RWR): RW is widely used in exploring graph structures in graph-related ML models (Perozzi et al., 2014; Nikolentzos & Vazirgiannis, 2020). RWR is closely associated with Personalized PageRank (PPR) (Page et al., 1999), which is popular for measuring node similarities. In Alg. 2, we give the implicit definition of $P(j|i, G)$ of RW and RWR by sampling. Note that for RW of window size $w$, we will perform the function $\texttt{RW}(i, G, \hat{w})$ for $w$ times, with $\hat{w} = 1, \cdots, w$.

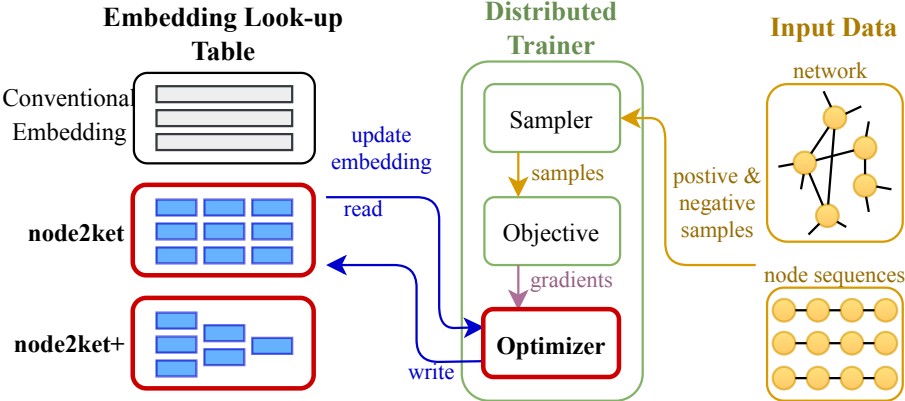

Figure 4: Considering such a common embedding learning pipeline that iteratively samples node pairs (yellow arrows), backward propagates gradients (the purple arrow), and updates node embeddings (blue arrows), we highlight our contributions in red including compressive embedding look-up tables by node2ket and node2ket+, and an optimizer to efficiently learn embeddings in a stable manner.

## C.2 OBJECTIVE FUNCTIONS

The basic idea for objective design is in accordance with the prevalent contrastive learning (Gutmann & Hyvärinen, 2010; Chen et al., 2020), making proximate nodes close in the embedding space and unrelated nodes far away.

**Skip-Gram with Negative Sampling (SGNS).** Initially, the objective is to maximize the following log softmax function:

$$O(i, j) = \log \frac{\exp(\langle \mathbf{x}_i, \mathbf{x}_j \rangle)}{\sum_{j' \in \mathcal{V}} \exp(\langle \mathbf{x}_i, \mathbf{x}_{j'} \rangle)}, \tag{7}$$

which, however, is computationally intractable for a large $\mathcal{V}$, and thus surrogate objectives are proposed, including Hierarchical Softmax and Negative Sampling (Mikolov et al., 2013b). In this paper, we only discuss a typical objective based on the popular Negative Sampling in NE literature:

$$O(i, j) = \underbrace{\log \sigma(\mathbf{x}_i^\top \mathbf{x}_j)}_{\text{positive samples}} + \underbrace{\sum_{k=1}^{K} \mathbb{E}_{j' \sim P_{neg}} \log \sigma(-\mathbf{x}_i^\top \mathbf{x}_{j'})}_{\text{negative samples}}, \tag{8}$$

where $\sigma(x) = 1/(1 + \exp(-x))$ is the sigmoid function, $K$ is the number of negative samples for each positive sampled pair, $P_{neg}$ is the distribution of negative samples $j'$ which is empirically set proportional to $\text{degree}(j')^{0.75}$ in SGNS-based NE methods (Perozzi et al., 2014; Tang et al., 2015; Grover & Leskovec, 2016; Liu et al., 2016).

**Gradient Derivation of SGNS.** Given the label $l$ ($l = 1$ for positive samples and $l = 0$ otherwise), the partial derivative of a logsigmoid function is:

$$\frac{\partial \log \sigma \big( 2(\mathbb{1}_l - 0.5) \langle \mathbf{x}_i, \mathbf{x}_j \rangle \big)}{\partial \mathbf{u}_{ic}} = \big( \mathbb{1}_l - \sigma(\langle \mathbf{x}_i, \mathbf{x}_j \rangle) \big) \frac{\partial \langle \mathbf{x}_i, \mathbf{x}_j \rangle}{\partial \mathbf{u}_{ic}}, \tag{9}$$

where $\mathbb{1}_l$ is an indicator returning 1 if $l = 1$ and 0 otherwise. The Eq. 9 is used in computing the gradient of SGNS loss.

**Marginal Triplets (MT).** The objective is more often used in knowledge graph embedding (Bordes et al., 2013; Zheng et al., 2020), word embedding (Meng et al., 2019), and also shows strong performance on structure-related tasks e.g. network alignment (Xiong et al., 2021). It aims to keep the margin between inner product positive pairs and negative pairs larger than a hyperparameter $\gamma$:

$$O(i, j) = -\max \Big( \langle \mathbf{x}_i, \mathbf{x}_{j'} \rangle - \langle \mathbf{x}_i, \mathbf{x}_j \rangle + \gamma, 0 \Big), \tag{10}$$

where $j'$ is the sampled negative node as $j'$ in SGNS. The gradient of the MT objective is obvious so we omit it here.

---

**Algorithm 2:** Sampling strategies $P(j|i, G)$.

---

**Input**: network $G$, starting node $i$, window size $w$ (for RW), restart probability $\alpha$;

**function** RW$(i, G, w)$

  1: $j \leftarrow i$;
  2: **for** $cw = 1, 2, \cdots, w$ **do**
  3:    $j \leftarrow$ SN$(j, G)$;                             ▶ Sample from $j$'s neighbors
  4: **end for**
  5: **return** $j$;

**function** RWR$(i, G, \alpha)$

  1: $j \leftarrow i$;
  2: **while** Rand$() < \alpha$ **do**
  3:    $j \leftarrow$ SN$(j, G)$;
  4: **end while**
  5: **return** $j$;

---

## D   THEORETICAL ANALYSIS

### D.1   COMPLEXITY ANALYSIS FROM THE PERSPECTIVE OF MODEL COMPRESSION

**Proposition 1** (Capability of Node2ket+ on Embedding Compression). *To train $q$-dimensional embeddings for $N$ nodes, if it requires obtaining $N$ different embeddings $\{\mathbf{x}_i\}_{i=1}^N$, by node2ket+, we need at least $C(Nq)^{1/C}$ parameters to store sub-embeddings and $NC$ parameters to store the index table $I(i, c)$, where $C$ is the number of sub-embeddings for each node.*

*Proof.* We use $|I(\cdot, c)|$ to denote the number of different indices of $c$-th sub-embeddings, then the total number of different embeddings that these sub-embeddings can constitute is $\prod_{c=1}^C |I(\cdot, c)|$. So we should make sure that $\prod_{c=1}^C |I(\cdot, c)| \geq N$.

The total number of sub-embeddings is $\sum_{c=1}^C |I(\cdot, c)|$:

$$\sum_{c=1}^C |I(\cdot, c)| \geq C\Big(\prod_{c=1}^C |I(\cdot, c)|\Big)^{1/C} \geq CN^{1/C}, \tag{11}$$

where the first '$\geq$' is by Cauchy inequality which takes '$=$' when $|I(\cdot, c)|$ is a constant. The dimension of each sub-embedding is $d = q^{1/C}$. So for the number of parameters for all the sub-embeddings is $d\sum_{c=1}^C |I(\cdot, c)| \geq C(Nq)^{1/C}$. The index table $I(i, c)$ needs $NC$ parameters. $\qquad\square$

### D.2   PROOFS TO THEOREM 1

Before the proof, we introduce the following lemma:

**Lemma 1** (Rank of Hadamard Product(Horn & Yang, 2020)). *Let $\mathbf{A}, \mathbf{B} \in \mathbb{R}^{N \times N}$ be positive semidefinite. If $\mathbf{A}$ and $\mathbf{B}$ have no zero main diagonal entries and*

$$\max(K\text{-}rank(\mathbf{A}) + rank(\mathbf{B}), rank(\mathbf{A}) + K\text{-}rank(\mathbf{B})) > N, \tag{12}$$

*then $\mathbf{A} \odot \mathbf{B}$ is positive definite.*

**Theorem 1.** *By node2ket, we denote $\overline{\mathbf{A}}_{(c)}(G) = \bigodot_{k \neq c} \mathbf{A}_{(k)}(G)$, we have $rank(\mathbf{X}^\top \mathbf{H}) = N$ if there $\exists c$ s.t. $\mathbf{A}_{(c)}(G)$ and $\overline{\mathbf{A}}_{(c)}(G)$ are positive semidefinite, and $\max\Big(K\text{-}rank\big(\mathbf{A}_{(c)}(G)\big) + rank\big(\overline{\mathbf{A}}_{(c)}(G)\big), rank\big(\mathbf{A}_{(c)}(G)\big) + K\text{-}rank\big(\overline{\mathbf{A}}_{(c)}(G)\big)\Big) > N$, and K-rank denotes the Kruskal rank.*

*Proof.* Notice that the conditions in the theorem meet the requirements of Lemma 1, so we have $\mathbf{A}(G) = \mathbf{A}_{(c)}(G) \odot \overline{\mathbf{A}}_{(c)}(G)$ is positive definite, which indicates that $rank(\mathbf{A}(G)) = N$. $\qquad\square$

## D.3 EXAMPLE 1 FOR TYPE-I EMBEDDING

**Example 1.** *For a undirected graph $G$ of 36 nodes ($N = 36$), whose adjacency matrix is defined as Fig. 5:*

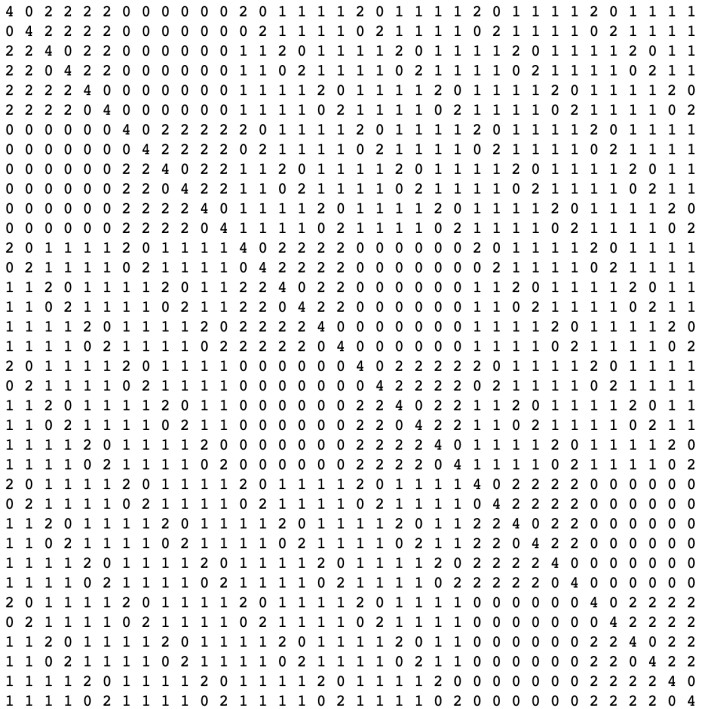

Figure 5: The adjacency matrix of the 36-node example.

*Suppose the embedding target is to preserve the adjacency matrix as the latent information matrix $\mathbf{A}(G)$. Given the number of parameters for each embedding $p = 8$, it can be factorized by node2ket without loss of information by setting:*

$$
\begin{aligned}
\mathbf{t}_1 &= [1, 1, 0, 0], \\
\mathbf{t}_2 &= [0, 0, 1, 1], \\
\mathbf{t}_3 &= [0, 1, 1, 0], \\
\mathbf{t}_4 &= [1, 0, 0, 1], \\
\mathbf{t}_5 &= [1, 0, 1, 0], \\
\mathbf{t}_6 &= [0, 1, 0, 1], \\
\mathbf{x}_i &= \mathbf{t}_{\lceil i/6 \rceil} \otimes \mathbf{t}_{i\%6}, i = 1, 2, \cdots 36,
\end{aligned}
\tag{13}
$$

*we would obtain that $\mathbf{X}^\top \mathbf{X} = \mathbf{A}(G)$ where $\mathbf{X} = [\mathbf{x}_1, \cdots, \mathbf{x}_{36}]$. Since $\mathbf{A}(G)$ is a rank-16 matrix, it can never be factorized as the multiplication of two $36 \times 8$ matrix by conventional embedding without losing information.*

### D.4 EXAMPLE 2 FOR TYPE-II EMBEDDING

**Example 2.** *For a directed ring $G$ of 9 nodes ($N = 9$), suppose we use the adjacency matrix as the latent information matrix $\mathbf{A}(G)$, i.e.*

$$\mathbf{A}(G) = \begin{bmatrix} & 1 & & & & & & & \\ & & 1 & & & & & & \\ & & & 1 & & & & & \\ & & & & 1 & & & & \\ & & & & & 1 & & & \\ & & & & & & 1 & & \\ & & & & & & & 1 & \\ & & & & & & & & 1 \\ 1 & & & & & & & & \end{bmatrix}, \tag{14}$$

*where the unfilled elements are 0 as the default. Given the number of parameters for each embedding $p = 6$, it can be factorized by node2ket without loss of information by setting:*

$$\begin{aligned} \mathbf{x}_1 &= [0,1,0] \otimes [1,0,0], \\ \mathbf{x}_2 &= [0,0,1] \otimes [1,0,0], \\ \mathbf{x}_3 &= [1,0,0] \otimes [0,1,0], \\ \mathbf{x}_4 &= [0,1,0] \otimes [0,1,0], \\ \mathbf{x}_5 &= [0,0,1] \otimes [0,1,0], \\ \mathbf{x}_6 &= [1,0,0] \otimes [0,0,1], \\ \mathbf{x}_7 &= [0,1,0] \otimes [0,0,1], \\ \mathbf{x}_8 &= [0,0,1] \otimes [0,0,1], \\ \mathbf{x}_9 &= [1,0,0] \otimes [1,0,0], \end{aligned} \tag{15}$$

*and*

$$\begin{aligned} \mathbf{h}_1 &= [1,0,0] \otimes [1,0,0], \\ \mathbf{h}_2 &= [0,1,0] \otimes [1,0,0], \\ \mathbf{h}_3 &= [0,0,1] \otimes [1,0,0], \\ \mathbf{h}_4 &= [1,0,0] \otimes [0,1,0], \\ \mathbf{h}_5 &= [0,1,0] \otimes [0,1,0], \\ \mathbf{h}_6 &= [0,0,1] \otimes [0,1,0], \\ \mathbf{h}_7 &= [1,0,0] \otimes [0,0,1], \\ \mathbf{h}_8 &= [0,1,0] \otimes [0,0,1], \\ \mathbf{h}_9 &= [0,0,1] \otimes [0,0,1], \end{aligned} \tag{16}$$

*we would obtain that $\mathbf{X}^\top \mathbf{H} = \mathbf{A}(G)$ where $\mathbf{X} = [\mathbf{x}_1, \cdots, \mathbf{x}_9]$ and $\mathbf{H} = [\mathbf{h}_1, \cdots, \mathbf{h}_9]$.*

*Since it is a rank-9 matrix, it can never be factorized as the multiplication of two $9 \times 6$ matrix without losing information.*

## E CONNECTIONS TO QUANTUM STATES

### E.1 PRELIMINARIES OF QUANTUM STATES

The quantum state is a physical concept to describe the state of a quantum system. It can be represented by a state vector, a wave function, or a density matrix. Paul Dirac introduced the **bra-ket notation** to denote quantum states as complex vectors in a Hilbert space (Dirac, 1939). In matrix representations, **kets** are column vectors denoted as $|\psi\rangle$ (where the name node2ket comes) and **bras** are row vectors that are the conjugate transpose of the kets denoted as $\langle\psi|$.

In physical experiments, the spin of an electron can be either up or down, therefore the quantum state of an electron is two-dimensional. A two-dimensional quantum system is known as a **qubit** (quantum bit), which is a basic unit in quantum computing. The state of a qubit is defined as a 2-dimensional complex vector in the **Hilbert space** $\mathbb{C}^2$. A state $|\psi\rangle = \begin{bmatrix} \alpha \\ \beta \end{bmatrix}$ is a linear superposition

of two orthonormal basis states $|0\rangle = \begin{bmatrix} 0 \\ 1 \end{bmatrix}$ and $|1\rangle = \begin{bmatrix} 1 \\ 0 \end{bmatrix}$, and we have $|\psi\rangle = \alpha|0\rangle + \beta|1\rangle$, where $\alpha, \beta \in \mathbb{C}$ are probability amplitudes satisfying $|\alpha|^2 + |\beta|^2 = 1$. That means, experimentally we will get the probability that $|\psi\rangle$ collapse to state $|0\rangle$ with probability $|\alpha|^2$ and state $|1\rangle$ with probability $|\beta|^2$. Indeed, any quantum state $|\psi\rangle$ should be **normalized**, i.e. the $l_2$-norm of state vector $|\psi\rangle$ should be 1, which indicates that the quantum fidelity $\langle\psi|\psi\rangle$ is equal to 1 in the language of quantum mechanics.

Considering two quantum systems $A$ and $B$ with corresponding Hilbert spaces $\mathcal{H}_A$ and $\mathcal{H}_B$, the composite space of $A$ and $B$ is $\mathcal{H} = \mathcal{H}_A \otimes \mathcal{H}_B$ where $\otimes$ denotes tensor product (can be seen as Kronecker product for two vectors). If we use $|\phi\rangle_A$ and $|\phi\rangle_B$ to denote states of system $A$ and $B$ respectively, then a state $|\phi\rangle \in \mathcal{H}$ that can be written as $|\phi\rangle = |\phi\rangle_A \otimes |\phi\rangle_B$ is called a **product state** or separable state, otherwise it is an **entangled state**. A most well-known entangled states is the Bell state $|\phi^+\rangle = \frac{1}{\sqrt{2}}(|0\rangle \otimes |0\rangle + |1\rangle \otimes |1\rangle) = (\frac{1}{\sqrt{2}}, 0, 0, \frac{1}{\sqrt{2}})^\top$. One can easily check that $|\phi^+\rangle$ cannot be written as $|\psi\rangle_A \otimes |\psi\rangle_B$ for any possible $|\psi\rangle_A, |\psi\rangle_B \in \mathbb{C}^2$. The properties of a 2-qubit system can be easily extended to an $n$-qubit quantum system that is in the Hilbert space $\mathbb{C}^{2^n}$.

### E.2 WHY WORD2KET CONSTRUCTS 'PSEUDO' ENTANGLED STATES?

In Sec. E.1 we have reviewed the concepts of quantum states, which is written in the language of physics. To transfer quantum states to embedding, we just need to substitute the complex number field $\mathbb{C}$ with the real number field $\mathbb{R}$ and quantum states $|\psi\rangle$ with embedding vectors $\mathbf{x}$ and $\mathbf{u}$.

Word2ket represent a node embedding $\mathbf{x}_i$ ($i$ is the node index) as:

$$\mathbf{x}_i = \sum_{r=1}^{R} \bigotimes_{c=1}^{C} \mathbf{u}_{irc}, \tag{17}$$

where $\mathbf{u}_{irc}$ is the state of a quantum subsystem. Though it is similar to the form of an entangled state, we find that it hardly follows the normalization rule of a quantum state, i.e. $\|\mathbf{x}_i\|_2 = 1$:

$$\|\mathbf{x}\|_2^2 = \sum_{r=1}^{R} \|\bigotimes_{c=1}^{C} \mathbf{u}_{irc}\|_2^2 + \sum_{r_1=1}^{R} \sum_{r_2=1, r_2 \neq r_1}^{R} \langle \bigotimes_{c=1}^{C} \mathbf{u}_{ir_1c}, \bigotimes_{c=1}^{C} \mathbf{u}_{ir_2c} \rangle. \tag{18}$$

To make $\|\mathbf{x}_i\|_2^2 = 1$, one feasible solution is to add constraint $\|\mathbf{u}_{irc}\| = 1/R$ and $\langle \mathbf{u}_{ir_1c}, \mathbf{u}_{ir_2c} \rangle = 0 \ \forall c, r_1 \neq r_2$. The latter constraint is too strict to be practical. Such constraints are not shown in word2ket in model training. Therefore, embeddings by word2ket are finally the so-called 'pseudo' entangled states.

### E.3 QUANTUM INTERPRETATION OF NODE2KET

The embedding constructed by node2ket can be regarded as the product state of several separate quantum sub-systems of which each state is an entangled state. We use a toy example to explain this: Suppose we define the embedding $\mathbf{x}_i$ as the vectorized 2-order tensor: $\mathbf{x}_i = \mathbf{u}_{i1} \otimes \mathbf{u}_{i2}$ where $\mathbf{u}_{i1}, \mathbf{u}_{i2} \in \mathbb{R}^8$, then the sub-embedding $\mathbf{u}_{ic}$ can be used to represent any state of a 3-qubit quantum system (since $2^3 = 9$) in the real number field, no matter it is a product state or an entangled state, and the embedding $\mathbf{x}_i$ is the product state composed of two separate quantum sub-systems represented by $\mathbf{u}_{i1}$ and $\mathbf{u}_{i2}$ respectively.

## F EXPERIMENTS

### F.1 EXPERIMENTAL SETTINGS

**Hardware Configurations.** All of the experiments are run on a single machine with Intel(R) Core(TM) i9-7920X CPU @ 2.90GHz with 24 logical cores and 128GB memory.

**Datasets.** We study five public real-world datasets of different scales, different densities, and containing different types of information. All of the networks are processed as undirected ones. The detailed statistics of the networks are given in Table 5. We give a brief introduction of the datasets as follows: **YouTube** (Tang & Liu, 2009b) is a social network with millions of users on youtube.com;

Table 5: Statistics of datasets.

| Dataset | $|\mathcal{V}|$ | $|\mathcal{E}|$ | Avg. degre | Type of info. | #Labels |
|---|---|---|---|---|---|
| PPI | 3,890 | 76,584 | 39.37 | Biological | 50 |
| GR-QC | 5,242 | 14,496 | 5.53 | Academic | N/A |
| BlogCatalog (BC) | 10,312 | 333,983 | 64.78 | Social | N/A |
| DBLP | 12,591 | 49,627 | 7.88 | Academic | N/A |
| YouTube-Cut (YTC) | 22,579 | 95,596 | 8.46 | Social | 47 |
| YouTube | 1,138,499 | 2,990,443 | 5.25 | Social | 47 |

**BlogCatalog** (Tang & Liu, 2009a) (BC) is a social network of bloggers who have social connections with each other; **PPI** (Breitkreutz et al., 2007) is a subgraph of the PPI network for Homo Sapiens; **Arxiv GR-QC** (Leskovec et al., 2007) is a collaboration network from arXiv and covers scientific collaborations between authors with papers submitted to General Relativity and Quantum Cosmology category; **DBLP** (Ley, 2002) is a citation network of DBLP, where each node denotes a publication, and each edge represents a citation.

**Baselines and Used Parameters.**   The investigated NE methods include three based on training with SGNS objectives (LINE, node2vec, VERSE), two based on matrix factorization (ProNE and NetSMF), one based on hyperbolic embedding (HHNE), one based on graph partition (LouvainNE), and one based on graph neural networks (GraphSAGE). Note that no baselines in tensorized/compressive NE are provided since this is the first work in this line of research.  Brief introductions of baselines and used parameters are as follows:

**LINE**[2] (Tang et al., 2015) models first-order and second-order proximity on the adjacency matrix, trains them separately with edge sampling, and concatenates the representations after normalization on node embedding. The number of samples is set as $1 \times 10^8$.

**node2vec**[3] (Grover & Leskovec, 2016) uses biased random walk to explore both breadth-first and depth-first structure, and train node embedding and hidden embedding by negative sampling. We set $p = 1$ and $q = 1$ in the experiments, and window size 10, walk length 80 and the number of walks for each node 80.

**VERSE**[4] (Tsitsulin et al., 2018) uses node similarity by Personalized PageRank (PPR), Adajcency Similarity, and SimRank, and learns node embedding by LINE-1st. We use the default version with PPR similarity, damping factor $\alpha = 0.85$. We run $10^5$ epochs for each node.

**NetSMF**[5] (Qiu et al., 2019) obtains node embeddings by sparse matrix factorization on a sparsified dense random walk matrix. We set the rank as 512, window size as 10, and rounds as 1000.

**ProNE**[6] (Zhang et al., 2019) initializes embedding by sparse matrix factorization, and then enhances it via spectral propagation. We run the Python version in the repository with the default settings. We use the enhanced embedding as the node embedding result.

**LouvainNE**[7] (Bhowmick et al., 2020) adopts Louvain graph partition algorithm (Blondel et al., 2008) and then learns node embeddings from the partitions. We set the damping parameter $a = 0.01$ of the method which is the default value in the repository.

**HHNE**[8] (Zhang et al., 2021) adopts hyperbolic embedding for heterogeneous networks. We feed random walk sequences into the hyperbolic embedding core. We set window size as 5, the number of negative samples as 10. The other parameters follow the default setting of the official repository.

---

[2]https://github.com/tangjianpku/LINE

[3]https://github.com/xgfs/node2vec-c

[4]https://github.com/xgfs/verse

[5]https://github.com/xptree/NetSMF

[6]https://github.com/THUDM/ProNE

[7]https://github.com/maxdan94/LouvainNE

[8]https://github.com/ydzhang-stormstout/HHNE

**GraphSAGE**[9] (Hamilton et al., 2017) To make the GNNs applicable to non-attributed networks, we adopt the same approach in (Hamilton et al., 2017) which treats the one hot encoding of node degree as input node features. We train the model in the unsupervised manner. We set the hidden dimension as 256, output dimension as 128, learning rate as 0.01 and rounds as 500. It should be noted that comparison of NE methods with GNN methods is not typical (Cui et al., 2018) because of their different settings: NE methods deal with networks with only structural information while GNNs deal with networks with node features.

**Work2ket** (Panahi et al., 2020) represent embeddings in the $R$-rank $C$-order CP Decomposition form with a neural network-based decoder. It cannot be directly run for NE, nor can the embeddings be optimized stably without the proposed optimization techniques in this paper. We name the variant that constructs embeddings like word2ket but optimized with techniques of node2ket as **w2k+n2k**. As the default, we set $R = 2$.

**Tasks.** Investigated tasks include the following three benchmark tasks in NE literature (Tang et al., 2015; Cao et al., 2015; Wang et al., 2016; Grover & Leskovec, 2016; Perozzi et al., 2014; Qiu et al., 2021): i) network reconstruction, preserving known links of the origin network, ii) link prediction, predicting unknown links in networks, and iii) node classification, predicting node labels by training an extra classifier on the learned node embeddings. The former two tasks can be categorized as structural tasks since they are only related to structural information. The node labels usually contain semantic information (Yu et al., 2022), so we categorize node classification as a semantic task. Detailed descriptions of the tasks and the metrics are as below.

**Network Reconstruction (NR).** Network reconstruction is used to measure an embedding model's ability to preserve nodes' origin local structure. In line with (Wang et al., 2016), for baseline methods, we measure the score of a pair of node embeddings by negative Euclidean distance. During the reconstruction process, we calculate the scores between all possible pairs of the embeddings, namely $|\mathcal{V}|(|\mathcal{V}| - 1)/2$ undirected node pairs. Then we sort the scores and select the top-$|\mathcal{E}|$ node pairs, denoted as $\mathcal{E}'$. Then the precision is defined as Precision $= |\mathcal{E} \cap \mathcal{E}'|/|\mathcal{E}|$. For large-scale networks, we use Prec@$N$ to approximately measure the precision.

**Link Prediction (LP).** We randomly mask 1% edges of the given network, and create the same number of edges as noise edges by randomly connecting nodes in the network. Then, we conduct network embedding on the network where masked edges are removed, and predict the masked edges from the mixture of masked edges and noise edges. The used metric is precision.

**Node Classification (NC).** As an experimental paradigm in line with previous works (Perozzi et al., 2014; Wang et al., 2016; Tang et al., 2015), we first train node embeddings unsupervisedly by utilizing merely structural information. For evaluation, we divide all the labels into train and test sets, and then we train an independent classifier to classify the nodes according to nodes' embeddings and labels in the training set. Then the trained classifier is used to classify the nodes in the test set by the learned node embeddings. Concretely, we conduct experiments on PPI and YouTube. After embeddings are obtained, LIBLINEAR (Fan et al., 2008) is used to train one-vs-rest Logistic Regression classifiers. We range the portion of training data from 1% to 9% for YouTube and 10% to 90% for PPI. We use Macro-F1 as the metric. For each embedding method, we perform 10 times of trials and report the average scores.

**Default Parameter Settings.** For all methods that can set the number of threads (node2ket and baselines except ProNE and LouvainNE), we use 8 threads as the default. For our method, we use the MT objective on the tasks of NR and LP, and the SGNS objective on the NC task. For the MT objective, the margin is set as $\gamma = 0.1$, and for the SGNS objective, the number of negative samples is set as $K = 5$. We use random walk with window size $w = 2$ for LP, $w = 1$ for NR and NC. The default Riemannian optimization order is 0. The default sampling number is 100M. For YouTube the sampling number is 1000M. Default input is data in the network form. Default embedding type is type-I.

---

[9]https://github.com/williamleif/GraphSAGE

Table 6: Network reconstruction precisions on medium-scale networks. 'CR.' is short for compressive ratio (see main text for the detailed definition).

|  | BC | DBLP | GR-QC | PPI | YTC |
|---|---|---|---|---|---|
| LINE | 4.80 | 2.34 | 54.19 | 15.18 | 0.10 |
| node2vec | 7.05 | 18.74 | 59.92 | 43.46 | 1.00 |
| VERSE | 10.96 | 16.96 | 56.08 | 27.59 | 7.26 |
| NetSMF | 7.64 | 18.08 | 30.26 | 17.14 | 0.28 |
| ProNE | 10.10 | 17.76 | 56.37 | 24.02 | 0.15 |
| LouvainNE | 6.79 | 21.33 | 43.47 | 17.77 | 11.90 |
| HHNE | 0.69 | 11.40 | 50.50 | 18.23 | <0.01 |
| GraphSAGE | 0.88 | 0.23 | 0.93 | 0.91 | 0.25 |
| w2k+n2k | 29.77 | 51.25 | 80.98 | 35.77 | 26.37 |
| node2ket | 50.83 | 88.97 | 97.90 | 81.56 | 74.62 |
| node2ket+ | **54.61** | **89.63** | **98.07** | **87.35** | **76.43** |
| CR. | 0.86 | 0.72 | 0.76 | 0.88 | 0.74 |

Table 7: Link prediction precisions on medium-scale networks. 'CR.' is short for the compressive ratio (see main text for the detailed definition).

|  | BC | DBLP | GR-QC | PPI | YTC |
|---|---|---|---|---|---|
| LINE | 53.08 | 75.45 | 95.86 | 55.15 | 62.76 |
| node2vec | 77.58 | 82.09 | 91.72 | 77.58 | 76.78 |
| VERSE | 33.35 | 68.81 | 93.79 | 51.55 | 59.83 |
| NetSMF | 59.34 | 82.09 | 87.59 | 72.16 | 74.58 |
| ProNE | 75.27 | 85.31 | 93.79 | 73.97 | 81.07 |
| LouvainNE | 57.99 | 73.44 | 86.90 | 57.99 | 82.11 |
| HHNE | 38.77 | 85.92 | 93.10 | 65.72 | 83.89 |
| GraphSAGE | 51.5 | 50.91 | 60.69 | 47.68 | <0.01 |
| w2k+n2k | 86.57 | 94.37 | 95.86 | 84.79 | 90.59 |
| node2ket | 89.28 | **95.37** | **96.55** | **84.54** | **91.63** |
| node2ket+ | **89.31** | 94.97 | 93.79 | **84.54** | **91.63** |
| CR. | 0.83 | 0.69 | 0.73 | 0.84 | 0.70 |

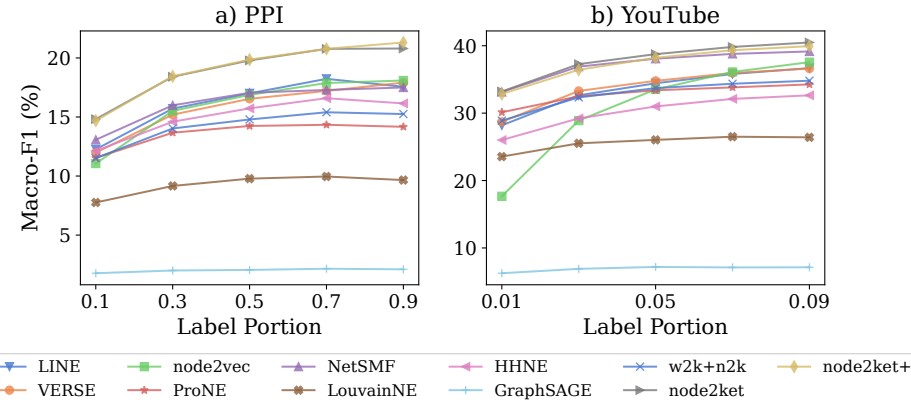

Figure 6: Results of node classification. The compressive ratio of node2ket+ is 0.90 on PPI, and 0.73 on YouTube.

## F.2 EXPERIMENTAL RESULTS COMPARED WITH CONVENTIONAL BASELINES

### F.2.1 RESULTS WITH THE SAME NUMBER OF PARAMETERS

We evaluate all the baselines and node2ket on the three tasks, NR, LP, and NC. In NR and LP, we set $p = 128$ for all methods, $C = 4$ and $R = 2$ for w2k+n2k, and $C = 8$ for our methods. In the experiments of NC, we set $p = 32$ for all methods, $C = 2$ and $R = 2$ for w2k+n2k, and $C = 4$ for our methods. For node2ket+, we conduct Louvain partition on the former 4 sub-embeddings with resolution [100, 500, 1000, 1500], and do not do partition on the later 4 sub-embeddings. The compressive ratio of node2ket+ in Table 6 and 7 is defined as the ratio of the number of parameters used by node2ket+ to that used by other methods.

**Analysis for Results.** We give the results of NR in Table 6, LP in Table 7, NC in Fig. 6. **i)** Our methods consistently outperform the baselines including conventional methods and w2k+n2k on all three tasks, demonstrating the superiority of our embedding construction in the high-dimensional Hilbert space compared with other low-dimensional NE methods with the same amount of parameters. **ii)** By comparing node2ket and node2ket+, we observe that the compressive node2ket+ shows better performance in NR, comparable in NC and LP. This indicates that the utilization of community information brings benefits to the preservation task NR and that appropriate compression won't have an adverse influence on model performance. **iii)** The embeddings learned by the GNN model GraphSAGE do not perform well on the three tasks. It is because GNN models highly rely on node features, which makes it difficult applying GNN methods on non-attributed networks.

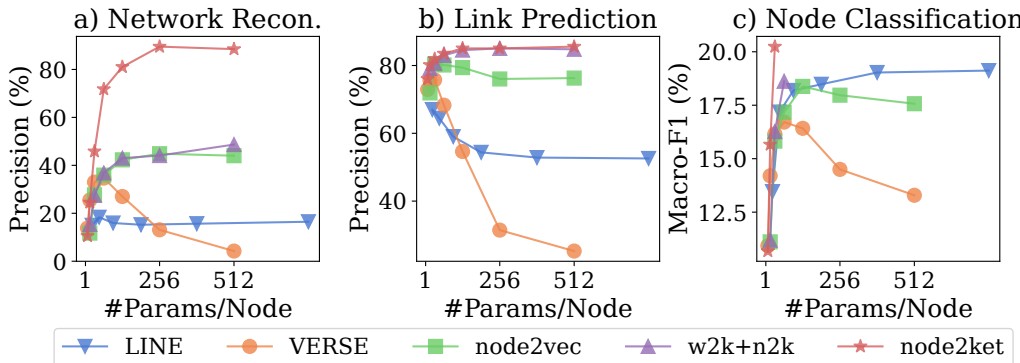

Figure 7: Results with varying parameters on PPI. The more close to the upper left corner, the better efficiency of parameter utilization.

### F.2.2 Results with Varying Training Parameters.

We evaluate the trainable baselines of which the used training parameters can be quantified. Experiments for NR, LP, and NC are conducted on the PPI. We give the results in Fig. 7, where #Params/Node (can be different from $p$ for conventional methods) for node2ket is equal to $Cd$. Specially, in Fig. 7 c), we only report the cases when #Params/Node$\leq 32$. That is because when #Params/Node$= Cd = 8 * 8 = 64$, the embedding dimension $q = d^C = 2^{24}$ exceeds the processing capacity of the classifier. In practice, the consumed memory is always positively related to the number of parameters (experiments in Sec. F.3)

**Analysis for Results. i)** In all the three tasks, node2ket achieves to use least parameters to yield the best results. Concretely, we observe that node2ket with only 32 parameters/node outperforms all the baselines with 512 parameters/node, which is 16 times as node2ket. The results indicate that our embedding models have better expressiveness. **ii)** The performance of conventional embedding methods is likely to get worse when the embedding dimensions are larger than a specific threshold. However, the phenomenon is not obvious on node2ket. The performance of node2ket improves almost consistently as more parameters are used, which indicates the better ability of node2ket in learning high-dimensional embeddings.

### F.3 Experiments on the Million-Scale Network

**Metrics. Network reconstruction.** For large-scale networks, it is infeasible to compute the scores of all $|\mathcal{V}|(|\mathcal{V}| - 1)$ node pairs and rank them to reconstruct the raw networks. So we approximately estimate the reconstruction precision by sampling in line with previous works e.g. SDNE (Wang et al., 2016). First we sample $(N - 1)|\mathcal{E}|$ randomly matched node pairs as the 'noise' node pairs, denoted as $\mathcal{E}^n$. Then we calculate the scores of all the $N|\mathcal{E}|$ edges in $\mathcal{E} \cup \mathcal{E}^n$ and select the top-$|\mathcal{E}|$ pairs as the reconstructed edges. So far, the computation of precision Prec@$N$ becomes obvious. The metrics of **link prediction** is the precision, the same as described in Sec. F.1; For **node classification** the metrics are Macro-/Micro-F1. In NC, we evaluate the embeddings with 1% portion of label information as the train data. **Settings.** We train 512-dimension embeddings for all methods. For node2ket and node2ket+, we set $C = 3$ and $p = 24$, which also yields 512-dimensional embeddings. For node2ket+, we do Louvain partition on the former 2 sub-embeddings with resolutions [500, 1000]. The comparison of model performance is given in Table 8. The comparison between the computational overhead is shown in Fig. 8. In Fig. 8 a), we plot the running time of the methods. For node2ket+, the time does not contain the procedure of building sub-embedding index table by Louvain partition which is regarded as data pre-processing and takes about 625 seconds on YouTube. In Fig. 8 b), we plot the maximum memory usage (VmPeak) which basically has three parts, the loaded data, temporary variables, and the node embeddings. In Fig. 8 c), we plot used local space for embedding storage. Embeddings of node2ket and other baselines are stored as '.npy' binary files. Sub-embeddings and the indices of sub-embeddings by node2ket+ are stored as pickle binary files. The methods 'node2ket(net)' and 'node2ket(seq)' in the results mean that node2ket uses network or node sequences by random walk as the input.

Table 8: Results on YouTube. The compressive ratio of node2ket+ is 0.02.

| Method | NR | | | LP | NC | |
|---|---|---|---|---|---|---|
| | Prec@2 | Prec@10 | Prec@50 | Precision | Macro-F1 | Micro-F1 |
| LINE | 81.17 | 66.40 | 55.95 | 55.27 | 15.75 | 29.77 |
| node2vec | 45.59 | 11.30 | 4.90 | 51.09 | 27.13 | 37.17 |
| VERSE | 37.32 | 26.45 | 22.91 | 23.77 | 12.82 | 23.04 |
| ProNE | 96.11 | 88.34 | 76.38 | 79.24 | **29.50** | 36.83 |
| LouvainNE | 85.65 | 70.50 | 62.92 | 76.06 | 18.57 | 31.06 |
| HHNE | 85.17 | 66.16 | 48.02 | 79.94 | 17.24 | 33.31 |
| NetSMF & GraphSAGE are not available on our machine | | | | | | |
| w2k+n2k(net) | **99.20** | **96.78** | **90.72** | **88.37** | 25.30 | 38.20 |
| node2ket(net) | 97.97 | 92.01 | 81.06 | 86.06 | **28.67** | **39.29** |
| node2ket(seq) | 98.02 | 91.67 | 78.03 | 86.22 | 25.71 | 37.44 |
| node2ket+(net) | 96.71 | 87.83 | 72.48 | 86.06 | 26.56 | 36.66 |
| node2ket+(seq) | 95.73 | 85.47 | 69.18 | 86.69 | 23.67 | 34.25 |

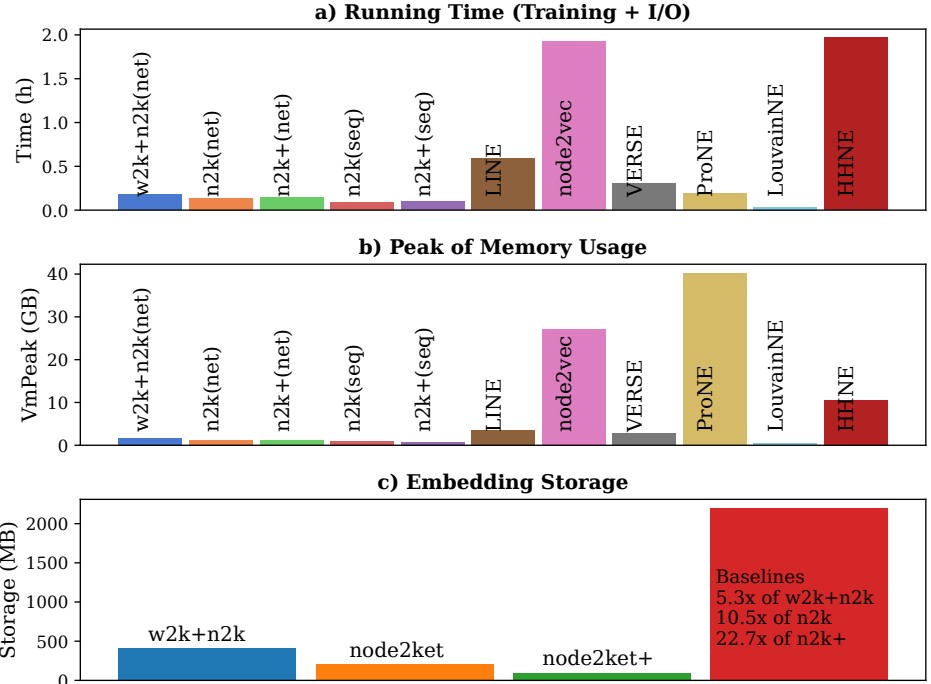

Figure 8: Computational overhead on YouTube, 'n2k' is short for 'node2ket'.

**Analysis. i)** Table 8 shows that our methods show overwhelming performance on structural tasks NR and LP, and comparable performance with the best baseline ProNE on the semantic task NC. **ii)** Fig. 8 shows that our methods use much fewer computational resources compared with baselines. It is the result of both much fewer parameters and the light-weight implementation of sampling. In our program, variables in memory are mostly the training sub-embeddings. In comparison, the baseline node2vec will store a huge transition matrix in memory. And ProNE, NetSMF will also consume a huge memory for large-scale matrix factorization. LouvainNE does not contain these parts so it is also highly light-weight. **iii)** Compared with node2ket, though sacrificing part of the model performance and slightly increasing the running time, node2ket+ moderately reduce memory usage and significantly reduce local storage while still outperforming most baselines. **iv)** Compared with the input of a network, the input of sequences can save much running time and memory usage, but sacrifices the local storage for node sequences since the file of node sequences is usually much larger than the network file.

### F.3.1   OVERHEAD COMPARISON WITH LIGHTNE AND GENSIM

**Settings. LightNE**[10] (Qiu et al., 2021) and **Gensim**[11] (Řehůřek & Sojka, 2010) are two CPU-only multi-threaded embedding tools with high efficiency for NE methods based on matrix factorization and word2vec respectively (See Sec. B for more details). We compare our node2ket (window size $w = 2$) with DeepWalk implemented with Gensim-4.3.1 and NetSMF integrated in LightNE on their ability to learn network structures. The thread number is set as 24 for all the methods. NR precisions are given in Table 9. The comparison of running time and the maximum memory usage (VmPeak) are shown in Fig. 9 a) and b) respectively.

**Result Analysis.** Results show that node2ket only needs **2.3GB** memory for 512-dimensional embedding for the YouTube network of 1.1M nodes. In comparison, the state-of-the-art high-efficiency NE system LightNE requires 94.0GB memory (**40.0x** of node2ket) and Gensim requires 17.5GB memory (**7.4x** of node2ket). Using much less memory, the precision of node2ket is only slightly lower than LightNE with about more 8 minutes of running time.

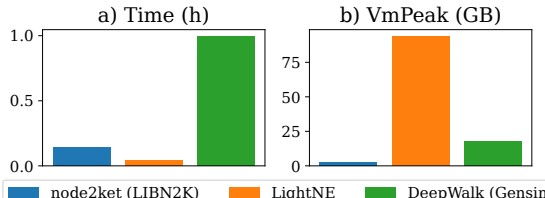

Figure 9: Overhead.

Table 9: NR precisions.

|  | Prec@N | | |
|---|---|---|---|
|  | N=2 | N=10 | N=50 |
| DeepWalk | 92.40 | 84.31 | 75.96 |
| LightNE | **98.78** | **97.60** | **94.35** |
| node2ket | 98.12 | 95.12 | 89.38 |

In experiments, we also find that memory usage will increase with the number of threads. We record VmPeak of our methods with 1 thread in Table 10. It shows that node2ket+ with sequences as input only consumes **0.32GB** memory. In comparison, LightNE (94.0GB) consumes **297.6x** of the memory and Gensim (17.5GB) consumes **55.2x** of the memory.

Table 10: VmPeak of our methods with 1 thread.

|  | n2k(net) | n2k+(net) | n2k(seq) | n2k+(seq) |
|---|---|---|---|---|
| VmPeak (GB) | 0.79 | 0.58 | 0.53 | 0.32 |

### F.4   ABLATION STUDIES

### F.4.1   MODEL PERFORMANCE WITH VARYING $C$ AND $p$

We conduct ablation studies with main concerns to parameters $C$ and $p$ on node2ket, while the conclusions are also suitable for node2ket+. We conduct experiments on the three tasks with varying $C$ and $p$. Results are given in Fig. 10. The analysis is from the following aspects:

**Model performance with the number of parameters** $p = Cd$ **fixed** ($d$ is the dimension of sub-embeddings). Blocks in each row in the sub-figures of Fig. 10 have the same $Cd$, indicating that the numbers of parameters are the same. We have the following findings: **i)** By observing the rows, we can find that in most cases, $C > 1$ is better than $C = 1$. So, we can safely claim that the strong performance of our methods as shown in Sec. F.2 comes from the tensorized embedding constitution in the Hilbert space with the same objective function, sampling strategy, and optimization technique. **ii)** Observing the rows, we can find that when $C$ is very big, though the compressive ratio is impressive, model performance decreases. It is probably because the embeddings become too sparse in the Hilbert space.

**Model performance with the embedding dimension** $q = d^C$ **fixed.** By the comparison between two cases of 64-dimensional embedding, $C = 1$, $p = Cd = 64$ and $C = 2$, $p = Cd = 16$ in the figures, we can find that the former will yield better results. It indicates that when the embedding dimension $q = d^C$ is fixed, though the embedding model can be compressed by increasing $C$, the model performance will also be sacrificed.

---

[10]https://github.com/xptree/LightNE
[11]https://github.com/RaRe-Technologies/gensim/tree/develop/gensim

**The compressive power of node2ket.** Impressively, as plotted in Fig. 10, the compressive ratio can be as small as $2^{-57}$.

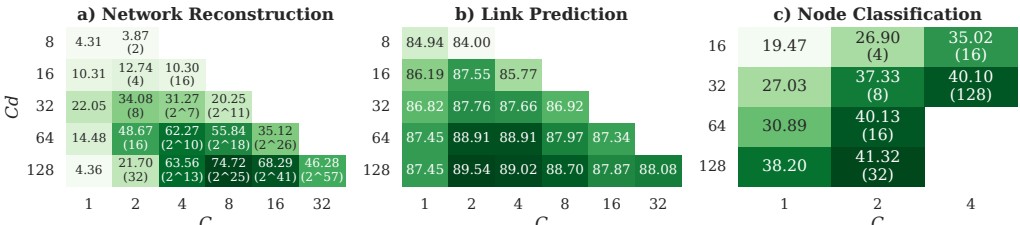

Figure 10: Model performance with varying $C$ and $p = Cd$. The number at the top of each block is the score, and the number at the bottom of each block within a bracket is the inverse number of the compressive ratio w.r.t. the conventional counterpart, which is calculated by $\frac{d^C}{Cd}$ – for example, $2^{10}$ means that the output embedding dimension $q = d^C$ is $2^{10}$ times as the used parameter $p = Cd$. The compressive ratios of b) are the same as a) so we hide them.

### F.4.2 ABLATION STUDY ON THE PROPOSED CONSTRAINTS

In Sec. 4.1, we introduced the two adopted constraints for embedding learning: i) sub-embeddings should be normalized, and ii) inner product between arbitrarily two sub-embeddings should be positive. We are going to investigate the necessity of the two constraints by removing them during embedding learning, and compare the model performance with different orders of Riemannian optimization whose algorithm is given in Alg. 1. We run experiments with random seeds varying from 1 to 5, then compute the average and std value of the results. The studies are conducted on PPI with other parameters and experimental settings keeping the same as in Sec. F.2. Results are presented in Table 11.

**Analysis for Constraint i).** Removing this will cause program crashes, as shown by the first row in Table 11. To learn embeddings with the constraint, we propose three orders of Riemannian optimization in Alg. 1, Sec. 4.3. We investigate how they affect the model performance, and present the results as shown by the bottom three rows of Table 11. The results show that Riemannian order=0 is recommended for its good performance and fast running speed.

**Analysis for Constraint ii).** We find that removing this constraint won't obviously affect the model performance by comparing the second row of Table 11 with the bottom three rows. However, by analyzing the sub-embeddings, we find that the phenomenon 'two negative makes a positive' happens 66,814 times without this constraint, which decreases to 24,414 times when we add the constraint. By visualizing the embeddings in Fig. 11, we also find that adding this constraint will endow the learned embeddings with better clustered local structure.

Table 11: Ablation studies for the adopted constraints on PPI. 'N/A' means that the program crashes.

|  | NR | LP | NC |
|---|---|---|---|
| Remove constraint i) | N/A | N/A | N/A |
| Remove constraint ii) | 87.87±0.07 | 84.48±0.46 | **16.54**±0.20 |
| Riemannian order=0 | **88.32**±0.07 | **84.79**±0.36 | 16.25±0.40 |
| Riemannian order=1 | 88.24±0.08 | 84.74±0.33 | 16.39±0.40 |
| Riemannian order=2 | 86.57±0.25 | 83.92±0.76 | 11.17±0.26 |

### F.4.3 ABLATION STUDY ON THE OPTIMIZER

In this paper we use Adagrad (Duchi et al., 2011) to adaptively modify the learning rate for the sake of stable learning with different sampling strategies and objectives. We compare the embedded Adagrad optimizer with other possible choices, including the SGD with a linearly decreasing learning rate from 0.02 to 0.0001, which empirically works best in training conventional embedding methods (Mikolov et al., 2013a; Perozzi et al., 2014; Tsitsulin et al., 2018; Grover & Leskovec, 2016; Xiong et al., 2022), and RMSProp by G. Hinton, which slows down the learning rate decaying of Adagrad.

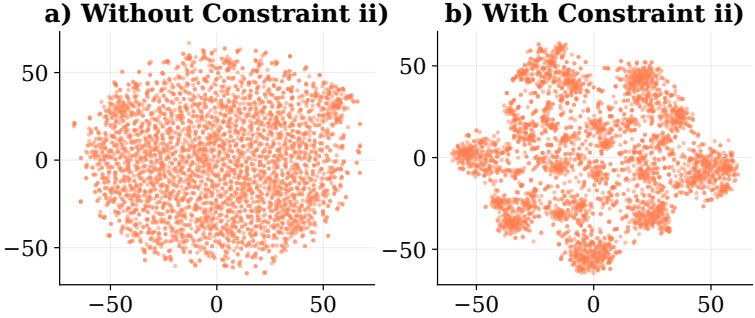

Figure 11: Visualization of learned node embeddings of PPI by t-SNE. The constraint ii) refers that we try to keep inner products between sub-embeddings positive. One can see that embeddings in b) are better clustered.

Results are given in Fig. 12 and Table 12, showing the superiority of Adagrad in both accelerating convergence and improving model performance on downstream tasks.

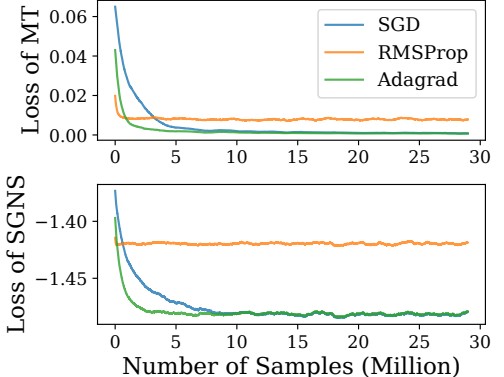

Table 12: Model performance with different optimizers on downstream tasks.

|  | NR | LP | NC |
|---|---|---|---|
| SGD | 48.58 | 88.18 | 38.42 |
| RMSProp | 7.94 | 82.01 | 30.52 |
| Adagrad | **76.43** | **91.63** | **38.96** |

Figure 12: Loss of objectives.

## F.5 MODEL PERFORMANCE IN DIFFERENT SCENARIOS

### F.5.1 SAMPLING STRATEGIES

We are investigating the model performance under different sampling strategies to see the model robustness. Experiments are conducted on YouTube-Cut. For NC, the embedding results are evaluated with 9% portion of all the labels as training data. We vary the window size $w$ for random walk, and vary the parameter $\alpha$ for random walk with restart. Results are given in Fig. 13. We can see that i) random walk with a small window size yields the best results, and ii) though the model performance fluctuates as the parameters $w$ and $\alpha$ change, the fluctuation is still in an acceptable range which can be seemed to be brought by the change of sampling strategies. Our carefully designed optimization techniques endow the embedding learning procedure with strong robustness when equipped with different sampling strategies.

### F.5.2 OBJECTIVES

We evaluate the model performance of node2ket on the dataset YouTube-Cut with two objectives, Marginal Triplets (MT) and Skip-Gram with Negative Sampling (SGNS), as described in Sec. C.2. The scores of node2ket on the three tasks NR, LP, and NC, are given in Table 13. NC uses 5% portion of train data. Results show that MT is more suitable for structural tasks NR and LP, while SGNS is more suitable for the semantic task NC. The conclusions are in accordance with previous works that use SGNS for NC task (Perozzi et al., 2014; Tsitsulin et al., 2018) and use MT for structural tasks (Xiong et al., 2021).

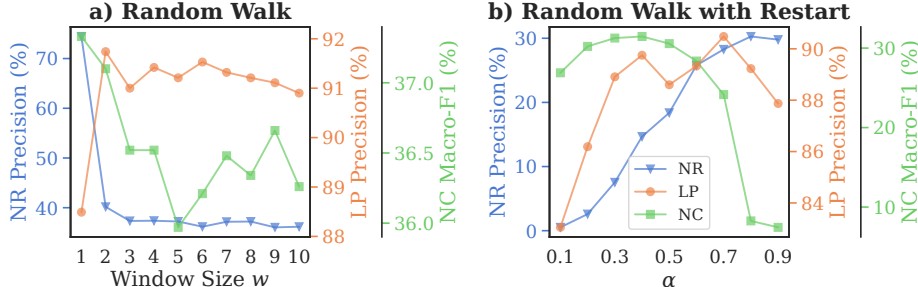

Figure 13: Performance of node2ket on YouTube-Cut under different sampling strategies.

Table 13: Scores of objectives MT and SGNS on the three tasks.

|  | NR (precision) | LP (precision) | NC (Macro-F1) |
|---|---|---|---|
| MT | **74.62** | **91.63** | 31.01 |
| SGNS | 0.32 | 87.13 | **38.96** |

### F.5.3 EMBEDDING TYPES

In Sec. 5.1 we have introduced two types of embedding architectures. We evaluate the model performance of node2ket on the dataset YouTube-Cut with the two embedding types. The scores of node2ket on the three tasks NR, LP, and NC, are given in Table 14. NC uses 5% portion of train data. Results show that the type-I embedding yields better results for node2ket and also demonstrate the robustness for different types of embedding.

Table 14: Scores of objectives MT and SGNS on the three tasks.

|  | NR (precision) | LP (precision) | NC (Macro-F1) |
|---|---|---|---|
| Type-I | **74.62** | **91.63** | **38.96** |
| Type-II | 18.29 | 88.91 | 37.06 |

### F.6 PARALLELIZABILITY

We evaluate the model performance and running time with regard to the number of threads on GR-QC. We set $C = 8$, $p = 128$. We plot the running time of 100M iterations and results of NR and LP with different numbers of threads in Fig. 14. We can see that: i) Used time decreases significantly as the thread number increases, indicating that parallelization can be applied to accelerate the embedding procedure. ii) The performance stays stable as the thread number increases, which indicates that asynchronously updating embeddings by multiple threads would not affect the model performance. iii) When the number of threads is greater than 8, the acceleration brought by parallelization becomes less significant, which means that the advantages of our method are more pronounced on machines with a limited number of threads.

### F.7 CASE STUDY BY INTERPRETABLE VISUALIZATION

In this section, we conduct visualization to show that how our methods preserve high-rank information with a small portion of parameters.

We conduct node embedding via node2ket on a circular ladder graph of 20 nodes, which is plotted in Fig. 15 a) and the adjacency matrix is visualized in Fig. 15 f). For node2ket, we set $C = 2$, and $p = 8$. After obtaining the sub-embeddings, we reconstruct the adjacency matrix from the $\mathbf{u}_{i1}$ and $\mathbf{u}_{i2}$, and visualize them in Fig. 15 b) and c). Then, from the final node embeddings, we compute the reconstructed adjacency matrix and visualize it in real-valued/binarized form as shown in Fig. 15 d) and e). From the results, we have the findings: i) by comparing Fig. 15 b), c), and d), we can see that column 1 and column 2 preserves structures in an approximately orthogonal manner, which means the extracted information is different among sub-embeddings, and ii) by comparing Fig. 15 e) with the ground truth f), node2ket fully reconstructs the input adjacency matrix.

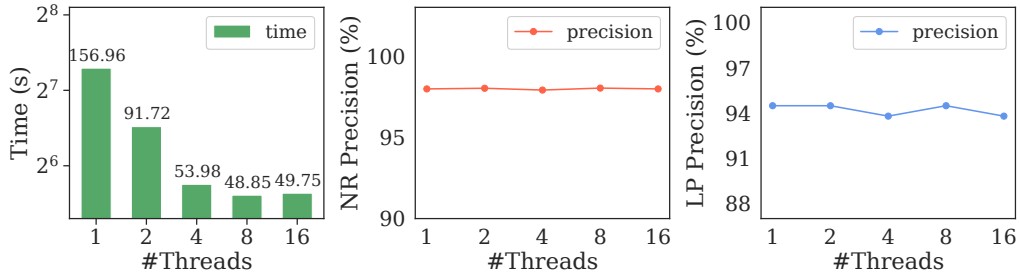

Figure 14: Parallelizability evaluation on the GR-QC dataset.

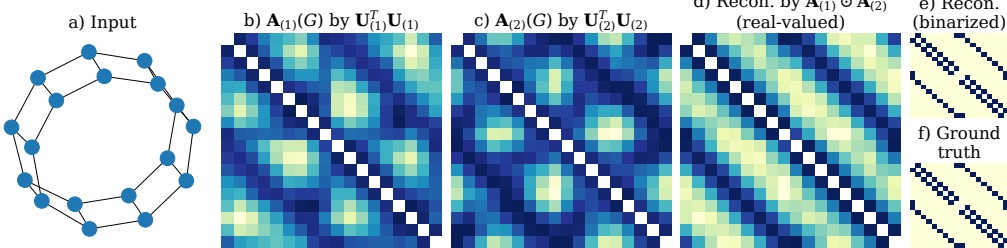

Figure 15: Visualization of the reconstructed adjacency matrix at different levels from the input 18-node circular ladder graph. Darker means a higher value. In b) and c), a color block at the $i$-th row and $j$-th column shows the inner product of two sub-embeddings, i.e. $\langle \mathbf{u}_{ic}, \mathbf{u}_{jc} \rangle$. And the heatmap becomes the visualization of $\mathbf{A}_{(c)}(G) = \mathbf{U}_{(c)}^T \mathbf{U}_{(c)}$. In d), a color block at the $i$-th row and $j$-th column visualizes the inner product of node embeddings, i.e. $\langle \mathbf{x}_i, \mathbf{x}_j \rangle$. In e), we binarize the results given by d). And f) is the visualization of the ground-truth adjacency matrix of the input graph.

To compare the reconstruction performance of different methods on different data, we further investigate a Hypercube graph, 3D-Grid graph, and the real-world Karate Club graph in Fig. 16. The results directly show the overwhelming performance of node2ket on preserving information especially the high-rank information.

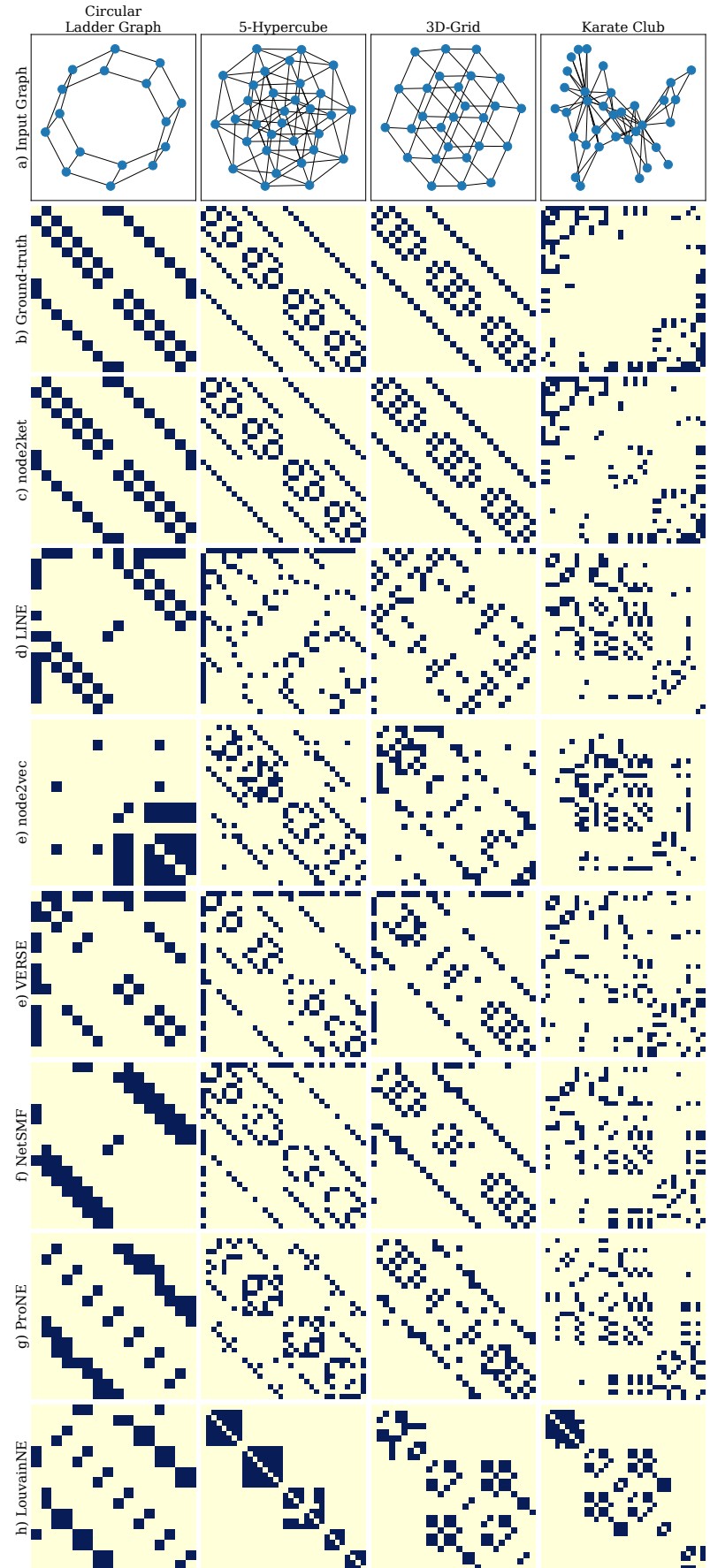

Figure 16: Visualization of reconstruction results of different embedding methods..

