# OpenReview forum: "Node2ket: Efficient High-Dimensional Network Embedding in Quantum Hilbert Space"
_ICLR.cc/2024/Conference — ICLR 2024 poster_

### Official Review · Reviewer_8cxP · 2023-10-25

**Soundness:** 3 good
**Presentation:** 3 good
**Contribution:** 3 good
**Rating:** 6
**Confidence:** 2

**Summary:**

This work proposes two high-dimensional network embedding methods called node2ket and node2ket+ that outperform standard methods such as word2ket. As evidence of these claims, they perform experiments studying several tasks, obtaining better compressive ratio than other approaches. Additionally, they provide an implementation in a library called LIBN2K.

**Strengths:**

- This paper improves upon existing network embedding techniques by proposing a high-dimensional embedding with better efficiency over previous quantum-inspired methods such as word2ket.
- The theoretical analysis is clear, showing that node2ket gets a high-rank approximation of the information matrix
- The experiments appear to be quite thorough and shows advantages over existing methods, as promised.
- It is nice that the code is made available.

**Weaknesses:**

- This is a quantum-inspired algorithm for classical machine learning, which suffers from the lack of a clear connection to quantum computation (see Questions).
- There are gaps for actually making this algorithm "quantum-friendly," as it is generally not easy to load classical information into a quantum device.

**Questions:**

- What is the significance of the embeddings being designed for "quantum Hilbert space"? To me, the relationship to quantum computation is not clear and seems more like an afterthought. Quantum computers are known to be good at problems with certain structure, and it's not clear to me what structure is being leveraged here (and what benefits are obtained as a result).
- The fact that pure quantum states are normalized leads to the constraint $\\|\mathbf{x}_i\\|=1$. Is there any consideration for eliminating a global phase, which I assume would affect the embedding? Also, I wonder what embedding might be developed for more general quantum states, such as mixed states.
- Forgive my ignorance, but what is the definition of positive node pairs and negative nodes? Does it just mean the inner product is positive or negative?

---

> ### Author Response · Authors · 2023-11-20
> **Response to reviewer 8cxP**
>
> Many thanks for your comments and appreciation. Your extensive knowledge in quantum computing will help us further improve our paper.
>
> ---
>
>
> ### **Q1: What is the significance of the embeddings being designed for "quantum Hilbert space"? To me, the relationship to quantum computation is not clear and seems more like an afterthought. Quantum computers are known to be good at problems with certain structures, and it's not clear to me what structure is being leveraged here (and what benefits are obtained as a result).**
>
> A1: Thanks for your feedback. We would like to emphasize that our approach is so far a classical one, aligning with the principles of quantum mechanics and potentials to be run on a quantum machine. We will provide a more detailed explanation below.
>
> We agree with the reviewer's observation that amplitude encoding theoretically allows the use of $n$ quantum bits to encode $q=p^n$ dimensional inputs, but often requires deep quantum circuits, where $p$ denotes the dimension of a single quantum bit. This implies an exponential increase in circuit depth with the input dimension $q$, demanding all quantum bits to be entangled and be well-connected physically [1].
>
> In contrast, in our paper, all quantum bits in the quantum system are organized into $C$ columns each containing $k$ qubits. **It is only required that each subsystem consisting of $k$ quantum bits has the capability to generate entanglement, while the $C$ subsystems are independent of each other.** Additionally, we appropriately configure parameters to ensure $q=(p^k)^C$, meaning we in fact use a subspace of the entire Hilbert space.
>
> As a result, the depth of the circuit for data loading is **independent of $q$ and only depends on $C$**, which is adjustable. It is possible to leverage circuit depth affordable by NISQ by restricting $C$ as a small value (e.g. 8 in our paper). We hold the view that this demonstrates the quantum-friendly in terms of data loading in our approach.
>
> [1] Plesch M, Brukner Č. Quantum-state preparation with universal gate decompositions[J]. Physical Review A, 2011, 83(3): 032302.
>
> ---
>
> ### **Q2: Is there any consideration for eliminating a global phase in the constraint $\|\mathbf{x}_i\|$, which I assume would affect the embedding?**
>
> A2: Though intuitively constraining $\|\mathbf{x}_i\|=1$ would erase part of information, **the erased information could be toxic for embedding learning**:
>
> As proved by [3], with the skip-gram embedding loss, as the training goes on, the embedding model will converge at:
>
> $$\lim\_{t\rightarrow \infty} \mathbf{x}\_i^{(t)\top} \mathbf{x}\_j^{(t)} = +\infty, \\\\ \lim\_{t\rightarrow \infty} \|\mathbf{x}\_i^{(t)} - \mathbf{x}\_j^{(t)}\| = 0.$$
>
> where the nodes i and j are a positive node pair, and t denotes the training iteration. The results lead to:
>
> $$\lim\_{t\rightarrow \infty} \|\mathbf{x}\_i^{(t)}\|^2 + \|\mathbf{x}\_i^{(t)}\|^2 = \lim_{t\rightarrow \infty} 2\mathbf{x}\_i^{(t)\top} \mathbf{x}\_j^{(t)} +\|\mathbf{x}\_i^{(t)} - \mathbf{x}\_j^{(t)}\|^2 = +\infty,$$
>
> which indicates that the modulus of some embeddings would go infinity. By constraining $\|\mathbf{x}\_i\|=1$, the above problem would fade automatically. Also, some works [2] show that the constraint empirically works.
>
> **Moreover, the most importantly, to make the node2ket program runnable, the constraint $\|\mathbf{x}\_i\|=1$ is empirially a must** as shown in the Table 11 (the row ''Remove constraint i)''), after removing which the program would crash.
>
> [2] Meng, Yu, et al. "Spherical text embedding." Advances in neural information processing systems 32 (2019).
>
> [3] H. Xiong, J. Yan and Z. Huang, "Learning Regularized Noise Contrastive Estimation for Robust Network Embedding," in IEEE Transactions on Knowledge and Data Engineering, vol. 35, no. 5, pp. 5017-5034, 1 May 2023, doi: 10.1109/TKDE.2022.3148284.
>
> ---
>
> ### **Q3: What embedding might be developed for more general quantum states, such as mixed states.**
>
> A3: As the quantum states go general, the difficulty of training them on classic machines will greatly increase. So far the scheme of representing embeddings by product state is the one that simultaneously can be run by classical machines with super efficiency and also can be well interpreted with quantum mechanics. A ''slightly'' entangled state might be more classical-friendly than a mixed state.
>
> Thanks for your suggestions and we would consider trying that on a real quantum machine.

---

> ### Author Response · Authors · 2023-11-20
> **Conti. page**
>
> ### **Q4: Definitions of ''positive node pairs and negative nodes''**
>
> A4: These are concepts from contrastive learning, which, given a batch of samples, aims to embed positive pairs of samples close but negative pairs of samples distant in the embedding space. In the embedding learning, such positive sample pairs are called ''positive node pairs'', and for a node in the node pair, we would sample some nodes (e.g. by random sampling from the whole networks) to form negative node pairs, and we call these nodes ''negative nodes''.
>
> Thanks for your suggestion and we will make it more clear in the updated version.
>
> ---
>
> Finally, hope you enjoy your Thanksgiving!

---

> > ### Comment · Reviewer_8cxP · 2023-11-22
> >
> > Thanks for the responses. I am certainly not an expert in this area, but I still feel the connection to quantum is lacking, and it doesn't make sense to call this quantum friendly (this is not something actually designed to be run on a quantum device). I'll keep the current score though, as this work seems interesting enough from the perspective of classical machine learning.

---

### Official Review · Reviewer_RkWJ · 2023-10-31

**Soundness:** 3 good
**Presentation:** 3 good
**Contribution:** 3 good
**Rating:** 6
**Confidence:** 2

**Summary:**

The paper introduces a groundbreaking paradigm for network embedding (NE), departing from traditional low-dimensional embeddings and exploring high-dimensional quantum state representations. The authors propose two NE methods, node2ket and node2ket+, and implement them in a flexible, efficient C++ library (LIBN2K). Experimental results showcase the good performance of their proposal, boasting advantages in parameter efficiency, running speed, and memory usage.

**Strengths:**

The paper is well organized. The insights offered in the paper have the potential to inspire the development of other quantum-inspired methods and contribute to the broader application of quantum computing in the field of network embedding.

**Weaknesses:**

The primary concern in this submission pertains to the technical contribution. First, the extension of the word2ket concept to product states appears relatively straightforward. Second, the utilization of product states might limit the embedding's expressivity since these states occupy a smaller portion of the Hilbert space and result in a low-dimensional representation.

**Questions:**

No questions at the moment.

---

> ### Author Response · Authors · 2023-11-20
> **Response to reviewer RkWJ**
>
> Many thanks for your comments and appreciation. Your suggestions are very valuable for the further works especially in the embedding theory.
>
> ---
>
> ### **Q1: The extension of the word2ket concept to product states appears relatively straightforward.**
>
> A1: It is true that node2ket and word2ket construct embeddings in a similar way. We believe that the embedding method approach created by word2ket is a great invention in the history of embedding research and would be a new paradigm for constructing embeddings in different fields. Compared to word2ket, our further contributions lie in the following aspects:
>
> - Node2ket is a successful trial of transferring word2ket from low-dimensional space to high-dimensional space, and from word embedding to node embedding (and possibly others with the toolkit LIBN2K).
>
> - We theoretically show the strong ability of the embedding construction as quantum states in preserving high-rank information.
>
> - By experiments, we show that representing nodes as product states is a more efficient and effective way to construct embeddings compared with the pseudo entangled states with the same amount of parameters.
>
> ---
>
> ### **Q2: The utilization of product state might limit model expressivity since these states occupy a smaller portion of the Hilbert space and result in a low-dimensional representation.**
>
> A2: i) ''The utilization of product state might limit model expressivity.'' It is true that, _**theoretically**_, product state (but not an entangled one) might limit model expressivity. However, so far, an efficient method to train embeddings as **strictly entangled states** has not been developed yet. We have compared the performance of product states (node2ket) and **pseudo entangled states** (w2k+n2k) in experiments, which shows that with the same number of parameters, embedding as product states is more efficient and effective compared with the pseudo entangled states.
>
> We will leave it as future works to explore how to train the strictly entangled states as embeddings efficiently and how well it performs in the tasks.
>
> ii) ''The product states occupy a smaller portion of the Hilbert space.'' We also consider this when studying the representation ability of the product quantum states. We want to know how much of the entire Hilbert space the product state occupies. However, it is a non-trivial mathematical problem to give an analytic result to answer the question. We only have an empirical conclusion: Although the product state does not occupy a significant portion of the entire Hilbert space, for any entangled state, there can always be a product state in its neighborhood that is very close to it. In fact, this question is very similar to asking how close the optimal solution of the rank-1 approximation by CP decomposition of an arbitrary tensor is to that tensor -- it is likely to be NP-hard.
>
> We also sincerely hope this question can be well answered by future research.
>
> ---
>
> Finally, hope you have a happy Thanksgiving!

---

> ### Comment · Reviewer_RkWJ · 2023-11-22
>
> Thank you for your rebuttal addressing my concerns. I will keep my score at 6.

---

### Official Review · Reviewer_VRn5 · 2023-10-31

**Soundness:** 3 good
**Presentation:** 3 good
**Contribution:** 3 good
**Rating:** 6
**Confidence:** 4

**Summary:**

For the standing and important task of network embedding in data mining and machine learning, the paper proposes explores the exponentially high embedding space for network embedding, which largely differs from existing works dewelling on the low-dimensional embedding. This is achieved by product quantum states in a super high-dimensional quantum Hilbert space. The experiments show surprisingly strong performance of the approach, in terms of both high memory and running efficiency with strong robustness across different tasks of network reconstruction, link prediction, and node classification. The authors also provide the source code to ensure the soundness of the experiments.

**Strengths:**

1) This paper innovatively resorts to the high-dimensional embedding space for network embedding, which quite departures from existing literature.
2) The paper is well presented and the overview plot in Fig. 1 is very informative and useful to readers. The paper is well organized with strong content in appendix that signifcantly enriches the paper.
3) The experiments are impressive. Provided with the source code, I am convinced by the strong performance.
4) The authors give strong theoretical understanding of the essence of their approaches, which I really appreciate.

**Weaknesses:**

As the authors emphasized, the presented techniques are mainly suited for the structure networks, without considering the attributes. I understand this setting and think it is reasonable in practice. It also fits with many previous works in literature that have also been compred in this paper.

**Questions:**

1) Comaperd with Fig. 1, can the authors provide a more succinct plot to convey the main idea of the paper? Fig. 1 is still a bit busy which is useful yet a more direct illustration in the begining of the paper is welcomed. Something like Fig. 2 is better.
2) Can the approach be useful for solving combinatorial problems especially for large-scale ones? As there is little attributes need to be considered in these cases thus it seems suited to the proposed methods?

---

> ### Author Response · Authors · 2023-11-20
> **Response to reviewer VRn5**
>
> Many thanks for your comments and appreciation. I feel so surprised to see that some points of your comments are exactly what I am doing right now or plan to give it a try recently. I am glad to exchange the ideas (maybe still preliminary) with you:
>
> ---
>
> ### **Q1: Potentials of being applied on attributed networks.**
>
> A1: Though the proposed node2ket cannot be directly applied to attributed networks, it does provide some insights in GNN model design. The common part of GNN and Network Embedding (NE) is that both of them aim to learn node representations, while one through an embedding look-up table (NE) and the other through neural networks (GNN). A typical application of our method in GNN is on the graph attention -- where the attention is conducted with the following two steps (simplified for illustration, not strictly the same as the literature):
>
> 1.compute the attention matrix $\mathbf{A}\in \mathbb{R}^{N \times N}$ by:
>
> $$\mathbf{A}=\mathbf{Q}\mathbf{K}=(\mathbf{W}\_Q\mathbf{X}\_{in})^T (\mathbf{W}\_K\mathbf{X}\_{in}),$$
>
> where the $\mathbf{Q},\mathbf{K}\in \mathbb{R}^{N\times d\_{hidden}}$ are named as the 'query' and the 'key' matrix respectively, $\mathbf{X}\_{in} \in \mathbb{R}^{d\_{in} \times N}$ is the input of the attention layer, and $\mathbf{W}\_Q, \mathbf{W}\_K \in \mathbb{R}^{d\_{hidden}\times d\_{in}}$ are the two weight matrices.
>
> 2.compute the output by:
>
> $$\mathbf{X}\_{out} = \mathbf{V} \textit{softmax} (\mathbf{A}) = (\mathbf{W}\_V \mathbf{X}\_{in})\textit{softmax} (\mathbf{A}),$$
>
> where the weight matrix $\mathbf{W}\_V\in\mathbb{R}^{d\_{out}\times d\_{in}}$ and the output $\mathbf{X}\_{out} \in \mathbb{R}^{d\_{out} \times N}$.
>
>
> We can find that the attention construction is exactly the same as the type-II Information Matrix Factorization (Eq. 5 in the paper) which is defined as follows:
>
> $$\mathbf{A}(G) \approx \mathbf{X}^T\mathbf{H},$$
>
> where $\mathbf{A}(G)$ is the latent information matrix, $\mathbf{X}, \mathbf{H} \in \mathbb{R}^{d\times N}$ are node embeddings and hidden embeddings respectively.
>
>
> **It is obvious that the attention $\mathbf{A}$ is a low-rank matrix whose rank $\textit{rank}(\mathbf{A}) \leq d_{hidden}$ just as the constructed latent information matrix $\textit{rank}(\mathbf{A}(G))\leq d$.** Though the low-rank bottleneck of attention has been pointed out in literature [1], and solutions to obtain high-rank attention also have been proposed in previous works e.g. [1,2], constructing the attention by representing the matrices $\mathbf{Q},\mathbf{K}$ in the way of embeddings by node2ket, i.e. representing $\mathbf{Q},\mathbf{K}$ by a Katri-Rao product (illustrated as the Fig. 2b in the paper):
>
> $$\mathbf{Q} = (\mathbf{W}_Q^{(1)}\mathbf{X}\_{in}) \circ (\mathbf{W}\_Q^{(2)}\mathbf{X}\_{in}) \circ \cdots \circ (\mathbf{W}\_Q^{(C)}\mathbf{X}\_{in})$$
>
> still has two strong advantages: i) **no extra parameters** are introduced, and ii) under certain conditions (Theorem 1 in the paper) the attention can be **full-rank**.
>
>
> [1] Bhojanapalli S, Yun C, Rawat A S, et al. Low-rank bottleneck in multi-head attention models[C]//International conference on machine learning. PMLR, 2020: 864-873.
>
> [2] Zhang Z, Shao N, Gao C, et al. Mixhead: Breaking the low-rank bottleneck in multi-head attention language models[J]. Knowledge-Based Systems, 2022, 240: 108075.
>
> ---
>
> ### **Q2: Potentials in solving CO problems.**
>
> A2: Indeed solving CO problems with the network embedding technique is what I am trying to do right now. For example, in solving the very large traveling salesman problem, especially in a non-Euclidean space (which means the distances are not derived from Euclidean coordinates), even the strongest heuristic LKH will face the challenge of a prohibitively large searching space. I believe that NE which has a long history in dimension reduction [3] will play a strong role in shrinking the search space.
>
> [3] Yan S, Xu D, Zhang B, et al. Graph embedding and extensions: A general framework for dimensionality reduction[J]. IEEE transactions on pattern analysis and machine intelligence, 2006, 29(1): 40-51.
>
> ---
>
> ### **Q3: Fig. 1 is still a bit busy.**
>
> A3: Thanks for the suggestion, we will modify the paper accordingly.
>
>
> ---
>
> In the last, I sincerely wish you a happy Thanksgiving!

---

### Author Response · Authors · 2023-11-20
**General response to all the reviewers**

Dear reviewers, thanks for your valuable time and comments.

The topic of high-dimensional embedding is so new that there are so many problems waiting to be solved and we can hardly solve them all in this single paper. Reviewers raised valuable questions and we believe these questions will point out the ways for the future development of this topic. We really appreciate your comments and suggestions!

And hope you have a wonderful Thanksgiving!

---

### Author Response · Authors · 2023-11-21
**Paper Updated**

Dear reviewers,

We have uploaded a new version of the paper, with former confusing words clarified (in red) and small typos fixed. If you have any further questions, we are very willing to continue the discussion and would greatly appreciate it if you kindly consider raising the score.

Thanks again!

---

### Meta-Review · Area_Chair_JQ4r · 2023-12-06

**Metareview:**

The paper proposes new quantum-inspired network embedding in very high dimensional space in contrast to existing network embedding in relatively low dimensional spaces. Previously such an embedding was developed for the embedding of words and the current paper extends the approach to the embedding of nodes in a network. The embedding uses product states to achieve a small representation in high dimensional space. The reviewers all appreciate the strong empirical performance compared with existing benchmarks.

On the other hand, one reviewer mentions that the new technique is for product states, which do not fully utilize the full high dimensional space. Furthermore, the current results mainly concern the network structure and do not take into account attributes. Last but not least, it should be noted that the technique is better referred to as a quantum-inspired algorithm for classical computers as opposed to quantum computers.

**Justification For Why Not Higher Score:**

All reviewers are positive about the paper but they also put it just above the acceptance threshold. The reviewers notes that the extension from the word embedding using quantum states to the current setting is not too hard.

**Justification For Why Not Lower Score:**

All reviewers are positive about the paper and appreciate the empirical performance.

---

### Decision · Program_Chairs · 2024-01-16

Accept (poster)